# A positive feedback loop between mTORC1 and cathelicidin promotes skin inflammation in rosacea

Zhili Deng[1,2,3,4,5,†], Mengting Chen[1,2,3,4,5,†], Yingzi Liu[1,2], San Xu[1,2,3,4], Yuyan Ouyang[1], Wei Shi[1], Dan Jian[1], Ben Wang[1], Fangfen Liu[1], Jinmao Li[1], Qian Shi[1], Qinqin Peng[1,4], Ke Sha[1,4], Wenqin Xiao[1,4], Tangxiele Liu[1,4], Yiya Zhang[1,2,3], Hongbing Zhang[6], Qian Wang[7], Lunquan Sun[1,3,4], Hongfu Xie[1,2,4,5] & Ji Li[1,2,3,4,5,8,*] (iD)

## Abstract

Rosacea is a chronic inflammatory skin disorder whose pathogenesis is unclear. Here, several lines of evidence were provided to demonstrate that mTORC1 signaling is hyperactivated in the skin, especially in the epidermis, of both rosacea patients and a mouse model of rosacea-like skin inflammation. Both mTORC1 deletion in epithelium and inhibition by its specific inhibitors can block the development of rosacea-like skin inflammation in LL37-induced rosacea-like mouse model. Conversely, hyperactivation of mTORC1 signaling aggravated rosacea-like features. Mechanistically, mTORC1 regulates cathelicidin through a positive feedback loop, in which cathelicidin LL37 activates mTORC1 signaling by binding to Toll-like receptor 2 (TLR2) and thus in turn increases the expression of cathelicidin itself in keratinocytes. Moreover, excess cathelicidin LL37 induces both NF-κB activation and disease-characteristic cytokine and chemokine production possibly via mTORC1 signaling. Topical application of rapamycin improved clinical symptoms in rosacea patients, suggesting mTORC1 inhibition can serve as a novel therapeutic avenue for rosacea.

**Keywords** LL37; mTOR; Rapamycin; rosacea; skin inflammation
**Subject Categories** Immunology; Skin

## Introduction

Rosacea is well established as a commonly chronic inflammatory skin disease featured by erythema, telangiectasia, papules, pustules, edema, or a combination of these symptoms, generally affecting the central face (Xie *et al*, 2017; van Zuuren, 2017; Deng *et al*, 2018). Epidemiologic studies of Caucasian populations showed that its prevalence has been 10% or higher (Steinhoff *et al*, 2011; van Zuuren, 2017). Although this cutaneous syndrome has been described centuries ago, the pathophysiological mechanisms remain ambiguous. Multiple therapies have been used for the management of rosacea, including oral tetracycline and isotretinoin, and topical application of azelaic acid, metronidazole, and vascular lasers (van Zuuren, 2017). However, due to specific therapeutic target has not been defined to be essential for the development of rosacea, most therapies are unsatisfactorily symptom-based treatments.

Epidemiological investigation suggests a genetic component for this disease, but no rosacea genes have been identified yet. Clinical and histopathological features indicate a dysregulation of cutaneous vascular, nervous, and immune systems in rosacea (Schwab *et al*, 2011; Steinhoff *et al*, 2011; Steinhoff *et al*, 2013). Though the knowledge about rosacea pathogenesis is still very limited, conventional consensus is that aberrantly excess production of cathelicidin is a hallmark of rosacea (Yamasaki *et al*, 2007; Yamasaki & Gallo, 2011). It has been recently assumed that responding to environmental rosacea triggers (such as heat, spicy food, UV, physical or chemical stimuli, and microbes), the pattern recognition receptors including Toll-like receptors (TLRs) are activated, followed by antimicrobial peptides (mainly the cathelicidin in the skin) or chemokine and cytokine production (Yamasaki *et al*, 2011; Buhl *et al*, 2015). Patients with rosacea have an increase in both cathelicidin and kallikrein 5 (KLK5, the serine protease that can cleave cathelicidin precursor protein into the active form), leading to the generation of the pro-inflammatory forms of antimicrobial peptide (e.g., LL37) (Yamasaki *et al*, 2007; Morizane *et al*, 2010). In addition to its antimicrobial

1 Department of Dermatology, Xiangya Hospital, Central South University, Changsha, China
2 Hunan Key Laboratory of Aging Biology, Xiangya Hospital, Central South University, Changsha, China
3 National Clinical Research Center for Geriatric Disorders, Xiangya Hospital, Central South University, Changsha, China
4 Key Laboratory of Molecular Radiation Oncology Hunan Province, Changsha, China
5 Key Laboratory of Organ Injury, Aging and Regenerative Medicine of Hunan Province, Central South University, Changsha, China
6 State Key Laboratory of Medical Molecular Biology, Department of Physiology, Institute of Basic Medical Sciences, Chinese Academy of Medical Sciences and Peking Union Medical College, Beijing, China
7 Hunan Binsis Biotechnology Co., Ltd, Changsha, China
8 Department of Dermatology, The Second Affiliated Hospital of Xinjiang Medical University, Urumqi, China
*Corresponding author. Tel: +86 731 84327472; E-mail: liji_xy@csu.edu.cn
†These authors contributed equally to this work

activity, LL37 can trigger a series of inflammatory processes (Schauber & Gallo, 2009; Gerber et al, 2011; Muto et al, 2014). In a mouse model, intradermal injection of human cathelicidin LL37 can induce an inflammatory response with rosacea-like phenotypes (Yamasaki et al, 2007). However, the regulatory mechanisms of abnormal over-production of cathelicidin LL37 and its subsequent actions remain largely undefined.

The mammalian target of rapamycin (mTOR) is a serine/threonine protein kinase involved in integrating a series of signals, ranging from nutrients and energy status to cellular stressors and cytokines, to regulate many fundamental cell processes in mammals (Laplante & Sabatini, 2012; Deng et al, 2015). mTOR binds with other proteins to form two different complexes, i.e., rapamycin-sensitive mTOR complex 1 (mTORC1) and rapamycin-insensitive mTOR complex 2 (mTORC2). mTORC1 consists of RAPTOR (regulatory-associated protein of mTOR), mLST8, PRAS40, DEPTOR, and mTOR. The heterodimer containing tuberous sclerosis 1 (TSC1) and TSC2 is a pivotal upstream regulator, which negatively regulates mTORC1 (Gan & DePinho, 2009; Russell et al, 2011). So far, protein synthesis is the best-identified process controlled by mTORC1. mTORC1 signaling phosphorylates S6 kinase 1 (S6K1) and the translational regulators eukaryotic translation initiation factor 4E (eIF4E)-binding protein 1 (4E-BP1), which in turn contribute to protein synthesis. In the immune system, mTOR signaling is emerging as an essential modulator of immune function due to its role in sensing and integrating signals from the surrounding microenvironment (Powell et al, 2012). Previous studies have demonstrated that mTOR signaling plays a roles in the pathogenesis of various cutaneous disorders, such as psoriasis (Buerger et al, 2013; Varshney & Saini, 2018) and atopic dermatitis (Naeem et al, 2017). Until now, however, there has been no proof of the direct connection between mTOR signaling and rosacea development.

Here, we focus on addressing the role of mTOR signaling in the pathogenesis of rosacea. We report that mTORC1 signaling is hyperactivated in both rosacea patients and mouse models. Functionally, both mTORC1 ablation and pharmacological inhibition by its specific inhibitors restrained the development of rosacea in an LL37-induced rosacea-like mouse model. On the contrary, hyperactivation of mTORC1 signaling in TSC2$^{+/-}$ mice exacerbated rosacea development. Furthermore, we revealed a positive feedback circuit between mTORC1 signaling and cathelicidin, in which LL37 activates mTORC1 signaling by binding to Toll-like receptor 2 (TLR2), which in turn enhances the expression of cathelicidin. Subsequently, cathelicidin LL37 derived from this loop stimulates NF-κB signaling and cytokine and chemokine production which are key factors associated with rosacea development probably through mTORC1 signaling. Finally, our pilot clinical study showed that topical application of rapamycin had a significant curative effect on rosacea patients. Collectively, these findings suggest a pivotal role for mTORC1 signaling in the pathogenesis of rosacea and reveal a potential therapeutic target for rosacea treatment.

## Results

### mTORC1 signaling is hyperactivated in rosacea

To explore the pathogenesis of rosacea, we first defined the transcriptional signatures of facial skin from patients with rosacea and healthy individuals (HS). To this end, we performed RNA-sequencing on lesional skin biopsies from the central face of 10 rosacea patients and normal skin biopsies from the similar site of 10 HS. We identified 4,417 differentially expressed genes (DEGs) between rosacea and HS ($P < 0.05$ and $|\log_2$ (fold change)$| > 1$), of which 2,149 genes were upregulated and 2,268 genes were downregulated in rosacea (Fig 1A and B; the entire list of DEGs is provided in Table EV1). Hierarchical clustering analysis demonstrated that rosacea skin samples can be significantly distinguished from HS (Fig 1A). Gene set enrichment analysis (GSEA) revealed 42 core KEGG terms ($P < 0.05$, FDR $< 0.25$) were enriched in rosacea (Table EV2). In consideration of the important role of the inflammatory response in the development of rosacea, we payed attention to the related KEGG pathways, among which mTOR signaling pathway was highlighted (Fig 1C and D), indicating that this pathway might be involved in the pathogenesis of rosacea.

To determine whether mTOR signaling plays a role in the development of rosacea, we detected the expression of pS6, the phosphorylated form of the mTORC1 downstream molecule S6 (Wendel et al, 2004), in skin samples from rosacea and HS, or patients with other facial inflammatory skin diseases of similar symptoms. Immunohistochemistry (IHC) results showed that all specimens from rosacea patients exhibited abundant cytoplasmic localization of pS6 in both epidermis and some infiltrating cells, whereas HS showed minimal activity of pS6 (Fig 1E and Appendix Fig S1A). Furthermore, our data showed a positive correlation between the pS6 expression levels in the epidermis and the Clinician's Global Severity (CGS) scores in rosacea patients (Fig 1F and Appendix Fig S1B). Immunoblot analysis also showed that mTORC1 signaling was dramatically upregulated in the lesional skin of rosacea patients, which was not directly correlated with an increase in mTOR protein levels (Appendix Fig S1C). Given that TSC1 and TSC2 are critical upstream negative regulators of mTORC1 (Russell et al, 2011), we detected their expression levels and showed that TSC2, rather than TSC1, was significantly decreased in rosacea (Appendix Fig S1D). Although pS6 expression was throughout the epidermis and some infiltrating cells in the dermis of rosacea patients, cytoplasmic localization of pS6 was only limited to superficial layers of the epidermis, hair sheath, and few infiltrating cells in the dermis or around the hair follicles in acne vulgaris or eosinophilic folliculitis skins (Appendix Fig S1E). Similarly, lupus erythematosus skin only showed mild pS6 expression in the epidermis and part of infiltrating cells (Appendix Fig S1F). In the meantime, we also detected the expression of p-Akt (Ser473), the phosphorylated form of Akt directly activated by mTORC2 through phosphorylating its hydrophobic motif (Ser473) (Sarbassov et al, 2005). By immunostaining, we demonstrated that only epidermis showed weak activity of p-Akt (Ser473), and there was no significant difference between rosacea and HS (Appendix Fig S1G).

Besides, we established a cathelicidin LL37-induced rosacea-like skin inflammation mouse model, which resembles the human disease phenotypes according to previously published studies (Yamasaki et al, 2007; Muto et al, 2014; Mascarenhas et al, 2017). As expected, the LL37-treated mice developed typical rosacea-like skin lesions with obvious clinical and pathological changes (Appendix Fig S1H). Consistent with the results in humans, mouse skin injected with LL37 showed an increase of pS6 in both epidermis and part of infiltrating cells compared with control mice (Fig 1G and

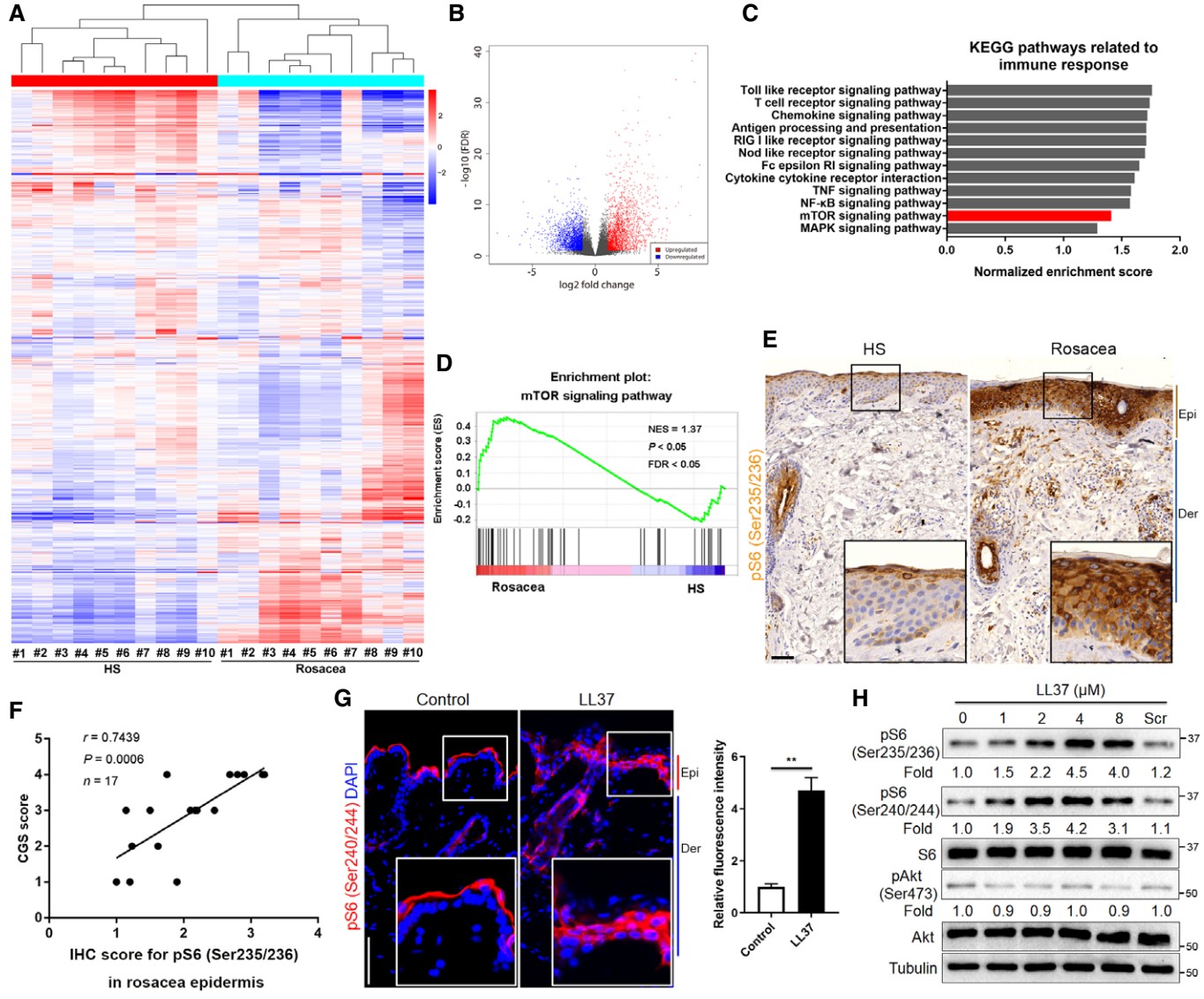

**Figure 1.  mTORC1 signaling is hyperactivated in skin lesions of rosacea.**

A   Heatmap of differentially regulated transcripts between HS and rosacea as determined by RNA-sequencing of whole skin lesions ($P < 0.05$). HS, skin biopsies from healthy individuals; Rosacea, skin biopsies from patients with rosacea. Blue color denotes low FPKM expression; red, high FPKM expression.

B   Volcano plot of differentially regulated genes between HS and rosacea. The red dots show the significantly upregulated genes; blue dots, significantly downregulated genes ($P < 0.05$).

C   Top-ranked upregulated KEGG terms related to immune responses in genes that were differentially regulated between rosacea and HS samples revealed through gene set enrichment analysis (GSEA).

D   GSEA on RNA-sequencing data from rosacea versus HS skin samples shows enrichment for mTOR signaling pathway in rosacea. NES, normal enrichment score; FDR, false discovery rate; FDR value < 0.05 was considered significant. Significance was calculated by permutation test.

E   Immunohistochemistry (IHC) of pS6 (Ser235/236) on skin sections from HS and rosacea. Higher magnified images of boxed areas are shown at the bottom right of lower magnified images for each group. Epi, epidermis; Der, dermis. Scale bar: 50 μm.

F   Correlation of pS6 (Ser235/236) expression (from IHC) in the epidermis of human rosacea lesions ($n = 17$) with the Clinician's Global Severity (CGS) scores. Spearman's correlation coefficient was used for the correlation analysis (two-tailed).

G   Immunostaining of pS6 (Ser240/244) in skin lesions from control and LL37-induced mice. Bottom right panels, magnified images of boxed areas. DAPI staining (blue) indicates nuclear localization. Scale bar: 50 μm. Right panel, the quantification of relative fluorescence intensity for pS6 (Ser240/244) in control and LL37 group ($n = 5$ for each group). Data represent the mean ± SEM. **$P < 0.01$. Two-tailed unpaired Student's $t$-test was used.

H   Immunoblotting of pS6 (Ser235/236 and Ser240/244), total S6, pAkt (Ser473), and total Akt in cell lysates from primary human keratinocytes treated with different doses of LL37 (0–8 μM) and scramble LL37 (8 μM) for 2 h. Scr, scramble LL37. pS6 and pAkt protein levels were analyzed relative to total S6 and Akt, respectively. Tubulin is the loading control. Data (H) are representative of at least three independent experiments.

Appendix Fig S1I), but there was no statistically significant difference in the expression of p-Akt (Ser473) between the two groups (Appendix Fig S1J). Since cathelicidin LL37 can activate mTORC1 signaling in epidermis of rosacea mouse model, we wondered whether it could stimulate mTORC1 in keratinocytes *in vitro*. By immunoblot, we demonstrated that LL37 activated mTORC1 signaling rather than mTORC2 signaling in a dose-dependent manner in primary human keratinocytes (Fig 1H). The above data suggested that mTORC2 signaling may not be involved in the development of rosacea. Thus, we focused on mTORC1 signaling thereafter.

Since external triggers, including heat and spicy food, can exacerbate the development of rosacea (Steinhoff *et al*, 2011; Gallo *et al*, 2018), we wondered whether these triggers could activate mTORC1 signaling in epithelial cells. To determine the effects of heat shock on mTORC1 signaling in primary human keratinocytes, we treated cells with heat shock (37, 42, 44 and 46°C) in the circulating water bath and detected the phosphorylation level of S6. To be mentioned, most of the cells died within 24 h at 46°C. By immunoblotting, we found heat shock (> 37°C) remarkably increased the phosphorylation level of S6 in (Appendix Fig S1K). Likewise, we found that mouse skin topically applied with capsaicin, another rosacea trigger factor, displayed hyperactivation of mTORC1 signaling in epithelial cells (Appendix Fig S1L).

Collectively, these data demonstrated that mTORC1, rather than mTORC2, is hyperactivated in the lesional skin of rosacea.

### Both genetic ablation and pharmacological inhibition of mTORC1 signaling block rosacea development

To determine the functional relevance of mTORC1 signaling in the development of rosacea, raptor was inducibly deleted in adult skin epithelium by crossing *raptor (fl/fl)*-floxed mice (Sengupta *et al*, 2010) with *K14-Cre^TM* mice (Sengupta *et al*, 2010). Offsprings from matings of *K14-Cre^TM/ raptor (fl/+)* mice yielded litters of the expected numbers, genotype, and mendelian ratio. Tamoxifen (TM) was intraperitoneally injected to induce cre-dependent recombination in *K14-Cre^TM/ raptor (fl/fl)* mice, which were indistinguishable from wild-type (WT) mice (also including *raptor (fl/+), raptor (fl/fl), K14-Cre^TM/ raptor (fl/+)*, and *K14-Cre^TM/ raptor (+/+)* mice). Six weeks after birth, the mice were intraperitoneally injected with TM for continuous 5 days. Ten days after TM injection, we shaved the mice and injected cathelicidin LL37 intradermally into WT and *K14-Cre^TM/raptor (fl/fl)* mice treated with TM (*Raptor* cKO; Fig 2A), and then compared the resulting rosacea-like features of *Raptor* cKO mice with age- and sex-matched WT mice treated with TM. By raptor and pS6 immunostaining, we confirmed that mTORC1 signaling was abolished in the epithelium of *Raptor* cKO mice (Fig 2B and Appendix Fig S2A). Our results showed that 12 h after last LL37 injection, WT mice exhibited obvious rosacea-like dermatitis, whereas *Raptor* cKO mice were unable to develop typical rosacea-like features (Fig 2C). Histological and RT–qPCR analysis showed that the inflammatory cell infiltration in the dermis and disease-characteristic factors were also dramatically improved in *Raptor* cKO mice compared with WT mice (Fig 2D–F).

In addition, we intraperitoneally injected mTORC1-specific inhibitor, rapamycin (RAPA), to investigate the inhibitory effect of RAPA on disease development (Appendix Fig S2B). As expected, RAPA significantly inhibited the activation of mTORC1 signaling (Appendix Fig S2C) and greatly improved rosacea-like dermatitis (Appendix Fig S2D). Also, the dermal infiltration and rosacea-associated genes expression were significantly reduced after RAPA treatment (Appendix Fig S2E–G). We confirmed our observations by using another mTOR inhibitor torin 1 (Francipane & Lagasse, 2013), which also showed an inhibitory effect on rosacea development (Appendix Fig S2H). Furthermore, we attempted to evaluate the potential of topical application of mTORC1 signaling-based therapy for rosacea. To this end, we prepared rapamycin ointment and topically applied it onto the injected site after LL37 injection (Fig 2G). We found that topical rapamycin treatment significantly improved the rosacea-like phenotypes and reduced infiltrating inflammatory cells in the dermis (Fig 2H–J). Taken together, these results demonstrated that mTORC1 deletion in epithelium or inhibition of mTORC1 signaling *in vivo* can prevent the pathological changes of rosacea and effectively alleviate disease development, suggesting a novel therapeutic strategy for rosacea.

### Hyperactivation of mTORC1 signaling aggravates the development of rosacea

To further clarify the role of mTORC1 signaling in rosacea development, we employed *TSC2* heterozygous (*TSC2^{+/−}*) mice, which

**Figure 2. Both conditional deletion and pharmacological inhibition of mTORC1 block rosacea development.**

A   Schematic diagram of intraperitoneal administration of tamoxifen (TM) for continuous 5 days before intradermal injection of LL37 in mice (*Raptor* cKO and WT mice). Mice were sacrificed on day 2 to conduct subsequent experiments. The mouse experiments were repeated for three times, and 5–8 mice were included in each group for each time. The results of a representative mouse experiment were displayed.
B   Immunostaining of Raptor in skin sections from WT and *Raptor* cKO mice. Epi, epidermis. Der, dermis. Scale bar: 50 μm.
C   The back skins of WT and *Raptor* cKO mice were intradermally injected with LL37 or control vehicle (*n* = 6 for each group). Images were taken 48 h after the first LL37 administration. Below panels, magnified pictures of yellow boxed areas. Scale bar: 2 mm.
D   HE staining of lesional skin sections from WT and *Raptor* cKO mice injected with LL37 or control vehicle (*n* = 6 for each group). Scale bar: 50 μm.
E   Dermal infiltrating cells were quantified (*n* = 6 for each group) for each high power field (HPF).
F   The mRNA expression levels of *Tnf-α, Il6, Mmp9,* and *Vegf* in skin lesions (*n* = 6 for each group).
G   Schematic diagram of topical application of rapamycin (RAPA) after LL37 injection in mice (BALB/c). Mice were sacrificed on day 2 to conduct subsequent experiments. The mouse experiments were repeated for three times, and 4–6 mice were included in each group for each time. The results of a representative mouse experiment were showed.
H   The back skins of LL37-injected mice topically applied with RAPA (*n* = 6 for each group). Images were taken 48 h after the first LL37 injection. Bottom panels, magnified images of yellow boxed areas. Scale bar: 2 mm.
I   HE staining of lesional skin sections from LL37 or control mice topically applied with RAPA or vehicle. Scale bar: 50 μm.
J   Dermal infiltrating cells were quantified (*n* = 6 for each group). Data represent the mean ± SEM. **P < 0.01. One-way ANOVA with Bonferroni's *post hoc* test was used.

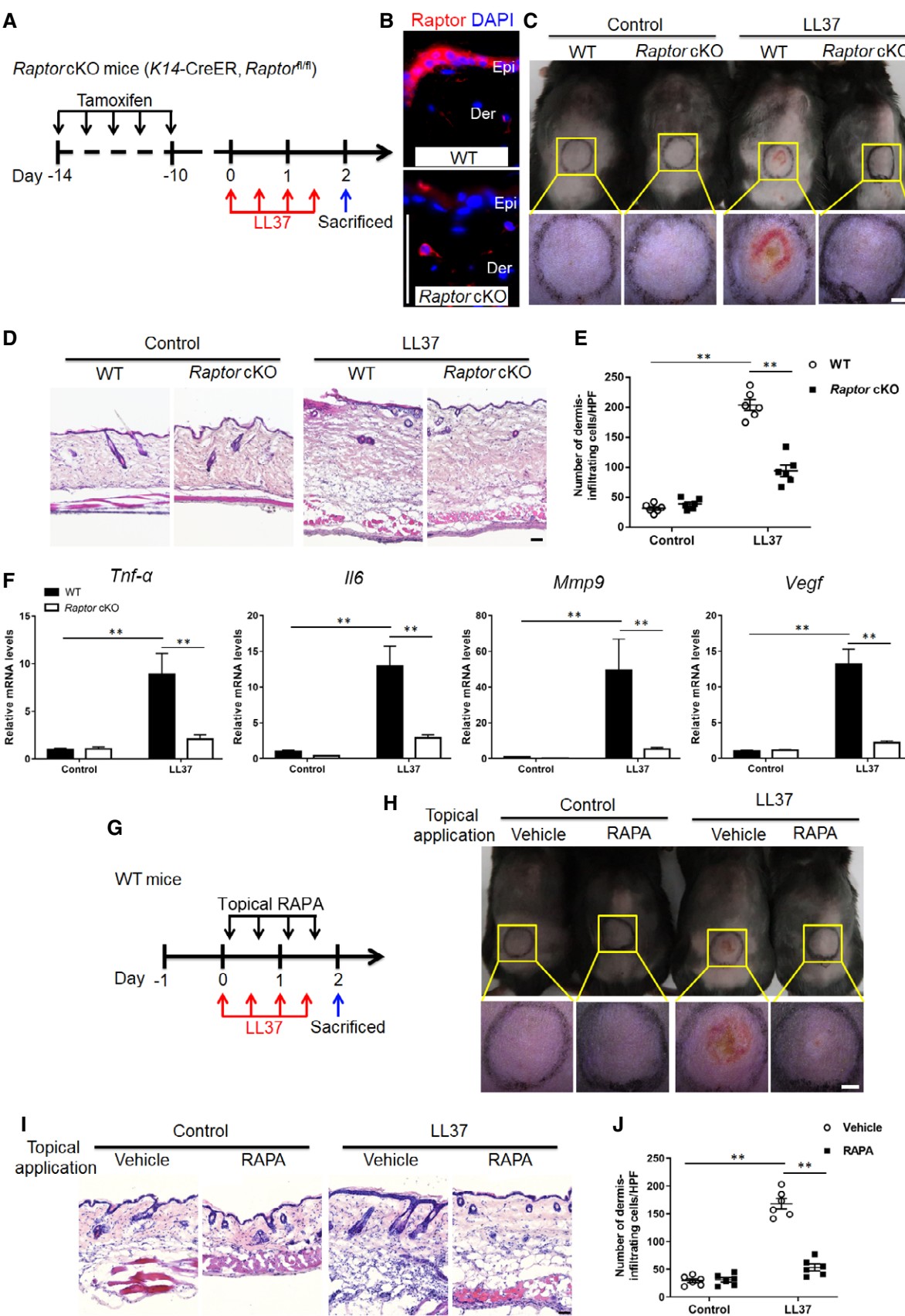

Figure 2.

exhibit mTORC1 hyperactivation compared with age- and sex-matched WT, namely $TSC2^{+/+}$ mice (Onda *et al*, 1999; Valvezan *et al*, 2017). We injected cathelicidin LL37 intradermally into $TSC2^{+/-}$ mice and compared the resulting inflammation with $TSC2^{+/+}$ (WT) mice. First, by pS6 immunostaining we verified that mTORC1 was indeed hyperactivated in $TSC2^{+/-}$ mice as well as in WT mice injected with LL37 (Appendix Fig S3). Our observation showed that 12 h after first injection, $TSC2^{+/-}$ mice had developed significant rosacea-like phenotypes, whereas $TSC2^{+/+}$ (WT) mice did not display obvious rosacea-like features until 48 h after the first injection which was consistent with previous description (Fig 3A). Moreover, the average redness area and score at 12 and 48 h were significantly increased in $TSC2^{+/-}$ mice (Fig 3B and C). After termination of LL37 injection, the rosacea-like dermatitis was remarkably improved in WT mice at day 8, but these clinical manifestations sustained in $TSC2^{+/-}$ mice until day 12 (Fig 3A–C). At day 2, the peak of rosacea-like changes, we found the dermal infiltrating cells were significantly increased in both $TSC2^{+/-}$ and WT mice injected with LL37, which was more evident in the dermis of $TSC2^{+/-}$ mice (Fig 3D and E). These results demonstrated that hyperactivation of mTORC1 signaling promotes and aggravates the development of rosacea.

## mTORC1 signaling regulates cathelicidin through a positive feedback circuit in keratinocytes

Abnormally increased expression of cathelicidin in the epidermis is a hallmark of rosacea (Yamasaki *et al*, 2007; Schwab *et al*, 2011; Yamasaki *et al*, 2011). To determine whether hyperactivation of mTORC1 signaling contributes to the over-production of cathelicidin in rosacea pathogenesis, we first confirmed the increased expression of cathelicidin in the lesional skin of rosacea patients at mRNA levels (Fig 4A). Then by immunostaining, we demonstrated that cathelicidin was highly expressed in rosacea and colocalized with increased pS6 in the epidermis. By contrast, cathelicidin and pS6 were much less abundant and only localized superficially in healthy skin (Fig 4B). Consistent with the results in humans, mice exposed to LL37 expressed significantly higher levels of cathelicidin in the epidermis, which was dramatically reversed in *Raptor* cKO mice (Fig 4C and D). Moreover, mTOR inhibitor, rapamycin, also attenuated the increased expression of cathelicidin in LL37-induced rosacea-like mouse skin (Appendix Fig S4A and B).

To further explore the regulatory relationship between cathelicidin LL37 and mTORC1 signaling, we treated primary human keratinocytes with different doses of LL37 (0, 1, 2, 4, 8 μM) and scrambled LL37 (8 μM). Intriguingly, consistent with the increased mTORC1 activity (Fig 1H), LL37 also stimulated the production of cathelicidin in a concentration-dependent manner (Fig 4E). These results indicate that LL37 may promote the expression of its precursor protein cathelicidin via activating mTORC1 in keratinocytes. To further verify this hypothesis, we pre-treated keratinocytes to abolish mTORC1 and then stimulated cells with LL37. Immunoblot analysis showed that inhibition of mTORC1 significantly suppressed the increased expression of cathelicidin after LL37 exposure at protein level (Fig 4F and Appendix Fig S4C), which was further confirmed by immunostaining (Fig 4G and Appendix Fig S4C). Collectively, these data reveal a positive feedback loop between mTORC1 signaling and cathelicidin in keratinocytes.

## Cathelicidin LL37 activates mTORC1 signaling by binding to Toll-like receptor 2 in keratinocytes

To gain insight into the molecular mechanisms by which LL37 activates mTORC1 signaling, keratinocytes were incubated with FITC-conjugated LL37 or scrambled LL37 and a red fluorescent dye, PKH26, which mainly binds to the cell membrane (Pasto *et al*, 2012). Then, the live cells were monitored using a fluorescence microscope. We found that LL37, rather than scrambled LL37, was localized to the cell membrane of keratinocytes (Fig 5A and B; Appendix Fig S5A and B), which reminded us that LL37 might activate mTORC1 through binding to the membrane receptors. TLR2, a membrane receptor expressed on the surface of certain cells and able to recognize foreign substances and transfer proper signals to the local cells, is aberrantly increased in the epidermis of rosacea patients (Yamasaki *et al*, 2011; Liu *et al*, 2014), which parallels the hyperactivation of mTORC1 signaling. Therefore, we speculated that LL37 might activate mTORC1 signaling via binding to TLR2. To verify this hypothesis, we performed lentiviral shRNA-mediated gene knockdown experiments to knockdown TLR2 and then treated keratinocytes with LL37. We found that LL37 routinely increased S6 phosphorylation, but this increase was abolished in TLR2-deficient cells (Fig 5C). On the contrary, the increased phosphorylation of S6 induced by LL37 was further enhanced in TLR2-overexpressing cells (Fig 5D). As expected, deficiency of TLR2 greatly abrogated the binding of LL37 to the live keratinocyte cells (Fig 5E and Appendix Fig S5C). To further investigate whether LL37 regulates mTORC1 activation by binding to TLR2, we performed co-immunoprecipitation (Co-IP) experiments. We first generated LL37 and scrambled LL37 both tagged with flag sequence (namely LL37-flag and sLL37-flag) *in vitro* (Appendix Fig S5D). Consistently, by immunoblot we showed that LL37-flag, rather than sLL37-flag, could activate mTORC1 signaling (Appendix Fig S5E). Then, the reciprocal immunoprecipitation analysis confirmed the interaction between LL37 and TLR2 (Fig 5F and G; Appendix Fig S5F and G). Taken together, these results support the hypothesis that cathelicidin LL37 activates mTORC1 signaling by binding to TLR2 in keratinocytes.

## Cathelicidin LL37 induces NF-κB activation and rosacea-associated chemokine and cytokine production

To further investigate the molecular mechanisms by which mTORC1 signaling initiates rosacea formation, we performed RNA-sequencing on skins from control and LL37-injected mice treated with or without RAPA. We identified 6,384 differentially expressed genes (DEGs) between LL37 and control mice (namely LL37 vs control), and 3,818 DEGs between LL37 mice and LL37 mice treated with RAPA (LL37 vs LL37 + RAPA; $P < 0.05$ and $|\log_2$ (fold change)$| > 1$; Fig 6A; Tables EV3 and EV4). There were a large number of DEGs ($n = 2,965$) shared between these two comparisons, indicating that inhibition of mTORC1 could significantly reverse the molecular changes in the skin of LL37-induced rosacea-like mouse model (Fig 6B). Gene set enrichment analysis (GSEA) revealed that NF-κB signaling pathway, which plays an important role in the regulation of chronic inflammation (Vallabhapurapu & Karin, 2009; Taniguchi & Karin, 2018), was the top-ranked KEGG term enriched in LL37-injected mouse skin, which was dramatically

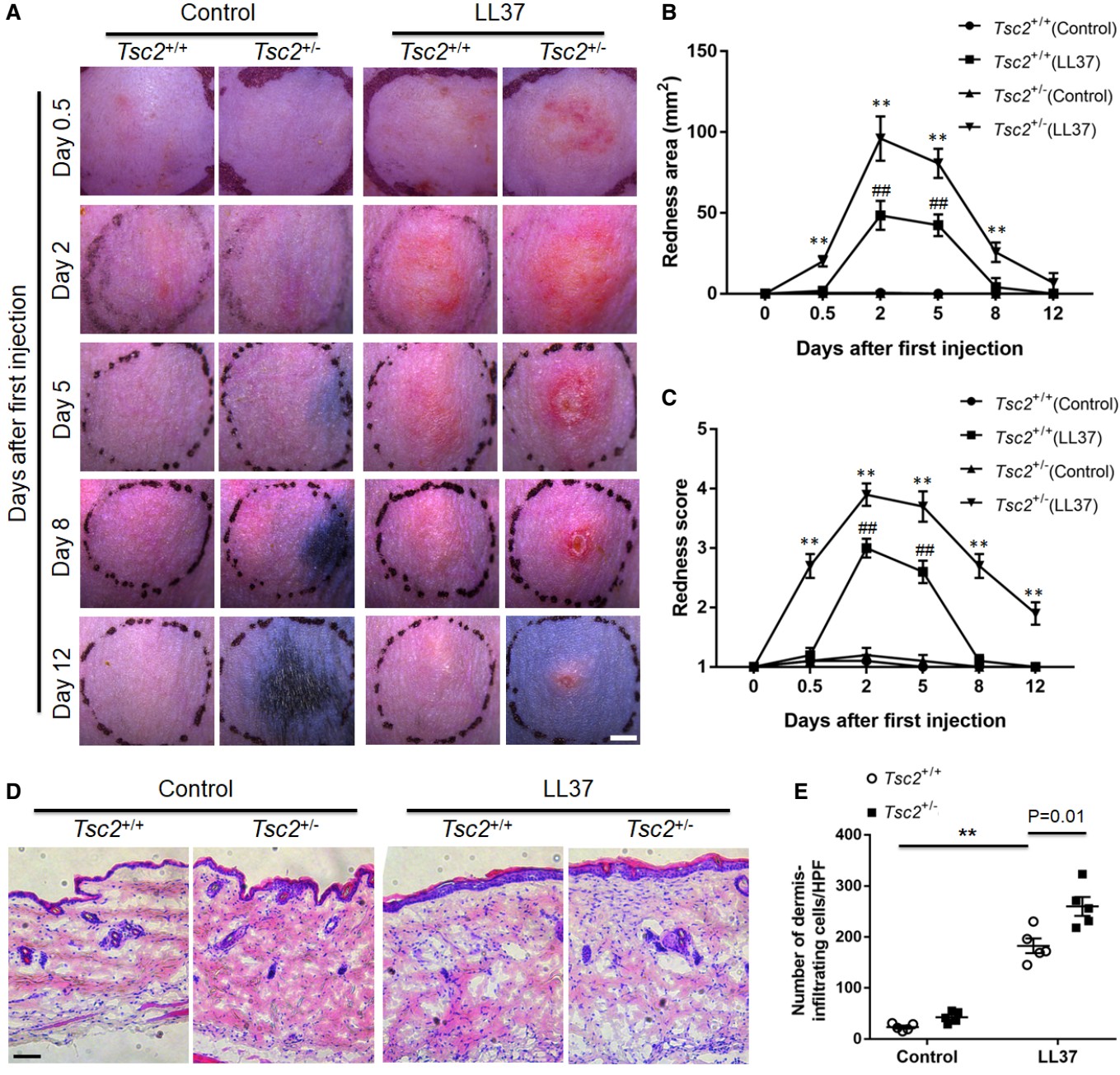

**Figure 3. Hyperactivation of mTORC1 accelerates rosacea development.**

A  The back skins of WT ($Tsc2^{+/+}$) and $Tsc2^{+/-}$ mice were intradermally injected with LL37 or control vehicle ($n = 5$ for each group). Pictures were taken on days 0.5, 2, 5, 8, and 12 after the first LL37 injection. The mouse experiments were repeated for three times, and five mice were included in each group for each time. The results of a representative mouse experiment were displayed. Scale bar: 2 mm.

B, C  The severity of the rosacea-like phenotypes was evaluated with the redness area (B) and score (C) ($n = 5$ for each group). **$P < 0.01$, comparison between $Tsc2^{+/-}$ (LL37) and $Tsc2^{+/+}$ (LL37) group. ##$P < 0.01$, comparison between $Tsc2^{+/+}$ (LL37) and $Tsc2^{+/+}$ (Control).

D  HE staining of lesional skin sections from $Tsc2^{+/+}$ and $Tsc2^{+/-}$ mice injected with LL37 or control vehicle ($n = 5$ for each group). Scale bar: 50 μm.

E  Dermal infiltrating cells were quantified ($n = 5$ for each group). Data represent the mean ± SEM. **$P < 0.01$. One-way ANOVA with Bonferroni's *post hoc* test was used.

suppressed via mTORC1 inhibition (Fig 6C and D; Tables EV5 and EV6).

To further determine whether mTORC1 regulates NF-κB, we detected the phosphorylation level of p65/NF-κB (p-p65) in the skin

of LL37-induced rosacea-like mice. By immunostaining, we found that cathelicidin LL37 injection significantly increased p-p65 in the epidermis, but this phenomenon was not obvious in *Raptor* cKO mice (Fig 6E and F). In addition, the NF-κB target genes (Il1α and

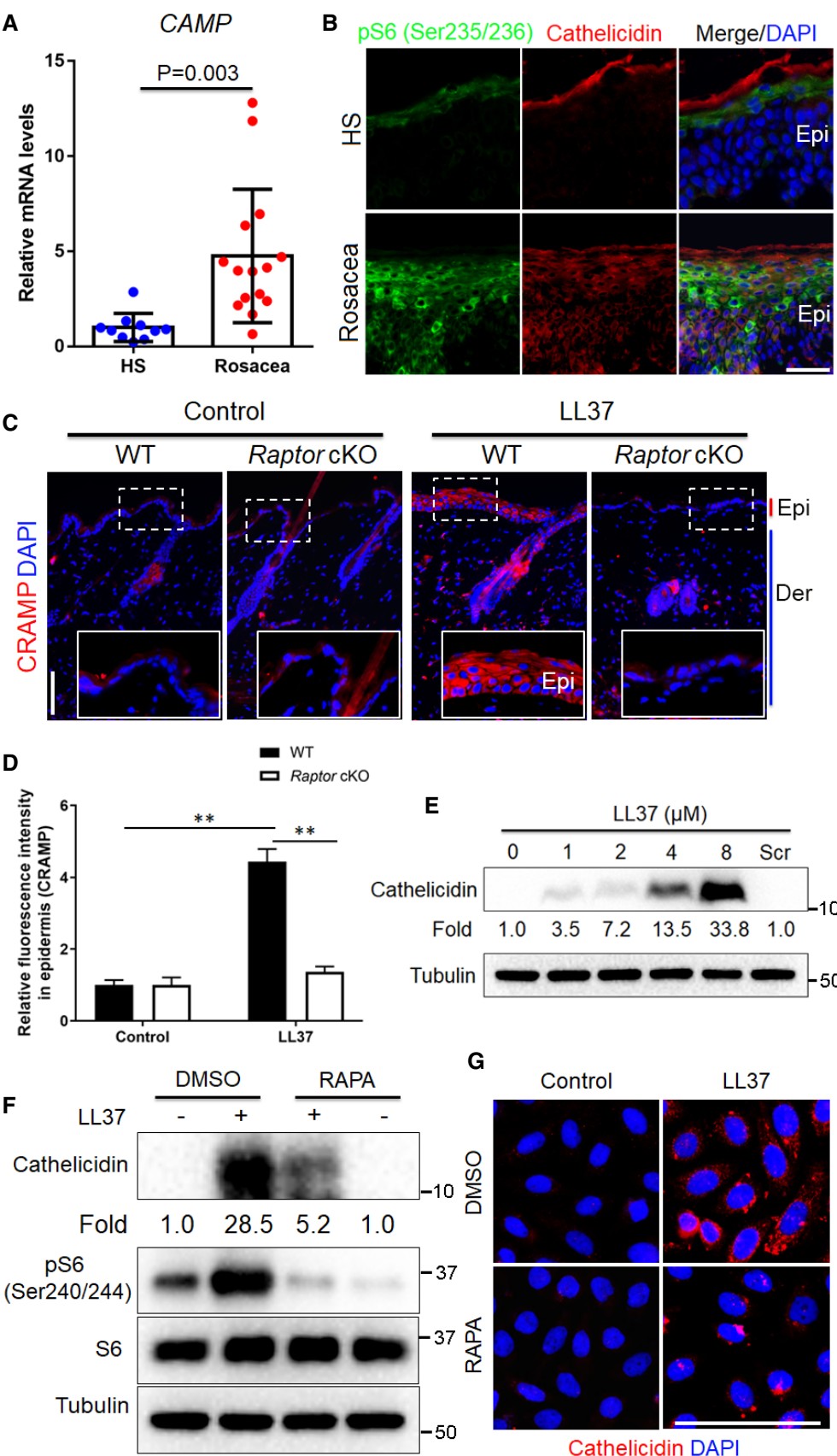

Figure 4.

**Figure 4.  mTORC1 regulates cathelicidin LL37 via a positive feedback loop in keratinocytes.**

A   The mRNA expression level of human cathelicidin (*CAMP*) in skin lesions from healthy individuals (*n* = 10) and patients with rosacea (*n* = 15).

B   Co-immunostaining of cathelicidin and pS6 on skin sections from HS and rosacea. Scale bar, 50 μm. Epi, epidermis.

C   Immunostaining of mouse cathelicidin (CRAMP) in skin sections from WT and *Raptor* cKO mice injected with LL37 or control vehicle (*n* = 6 for each group). Bottom right panels, magnified images of dotted line areas. Epi, epidermis; Der, dermis; Scale bar: 50 μm.

D   Quantification of relative fluorescence intensity for CRAMP in epidermis (*n* = 6).

E   Immunoblotting of cathelicidin in cell lysates from primary human keratinocytes treated with different doses of LL37 (0–8 μM) and scramble LL37 (8 μM) for 24 h. Cathelicidin protein levels were analyzed.

F   Immunoblot analysis of cathelicidin and pS6 in cell lysates from primary human keratinocytes treated with LL37 (4 μM) ± RAPA for 24 h. Cathelicidin protein levels were analyzed.

G   Immunostaining of cathelicidin in HaCaT keratinocytes treated with LL37 (4 μM) ± RAPA for 24 h. DAPI staining (blue) indicates nuclear localization. Scale bar: 50 μm. All results are representative of at least three independent experiments. Data represent the mean ± SEM. **$P$ < 0.01. Two-tailed unpaired Student's *t*-test (A) or 1-way ANOVA with Bonferroni's *post hoc* test (D) was used.

Il1β) and NF-κB family of transcription factors (including Nfκb1, Nfκb1, Rela, and Relb) (Hiscott *et al*, 1993; Mori & Prager, 1996) were upregulated in LL37-treated skin, whereas this upregulation was markedly suppressed in *Raptor* cKO mice (Fig 6G and Appendix Fig S6A). Similarly, inhibition of mTORC1 by rapamycin also abolished the activation of NF-κB signaling in the epidermis (Appendix Fig S6B and C). To investigate how mTORC1 activates NF-κB signaling, we detected the expression of TNF-α, one of the most potent inducers of NF-κB, in LL37-stimulated primary human keratinocytes treated with or without rapamycin. By qPCR and ELISA, we showed that LL37 promoted the production of TNF-α via mTORC1 signaling (Appendix Fig S6D).

To substantiate the notion that our results regarding NF-κB might have an application to human disease, we performed GSEA with RNA-sequencing data of rosacea patients and confirmed that a number of molecules associated with NF-κB were strongly increased in skin lesions of rosacea patients compared to specimens from HS (Fig 6H). By immunohistochemistry, we further verified that the nuclear localization of p65 was significantly increased (Fig 6I and J), indicating that NF-κB signaling was hyperactivated in the lesional skin of patients with rosacea.

Besides, GSEA also revealed that inhibition of mTORC1 signaling significantly declined the enrichment of chemokine signaling pathway in LL37-induced rosacea-like skin lesions (Fig 7A). The abnormal stimulation of rosacea-characteristic chemokines (Cxcl11, Cxcl12, Ccl2, and Ccl3) was dramatically remitted in *Raptor* cKO mouse skin (Fig 7B), and these findings were further confirmed in mice treated with rapamycin (Appendix Fig S7A). We further demonstrated that induction of chemokines by LL37 could be blocked via mTORC1 suppression in primary human keratinocytes *in vitro* (Fig 7C). Consistently, we found that chemokine signaling pathway is also enriched in the lesional skin of rosacea patients (Fig 7D and E; Appendix Fig S7B).

Collectively, these data suggest an important role for the mTORC1-NF-κB/chemokine axis in the pathogenesis of rosacea.

## Topical application of rapamycin improves rosacea-associated symptoms in rosacea patients

Since mTOR inhibitors (e.g., rapamycin and everolimus) have been approved by the U.S. Food and Drug Administration (FDA) for the treatment of facial angiofibromas in patients with tuberous sclerosis complex, and our above findings revealed the key role of mTORC1 signaling in the pathogenesis of rosacea, we wondered whether inhibition of mTORC1 was able to alleviate rosacea-associated symptoms in rosacea patients. Eighteen female patients diagnosed with rosacea were included in this assay and were randomized to receive either placebo (*n* = 8) or 0.4% FDA-approved rapamycin ointment (*n* = 10). No other treatment was undertaken within 1 month before and during topical rapamycin treatment. Rosacea skin lesions were topically applied with placebo or rapamycin ointment two times a day for four continuous weeks. After four weeks of topical rapamycin treatment, patients showed clinically improvement compared with placebo treatment (Fig 8A and B). CEA and IGA scores were both significantly decreased after topical rapamycin treatment (Fig 8C and D; Appendix Table S3). There were no obvious adverse



**Figure 5.  Cathelicidin LL37 activates mTORC1 through binding to TLR2 in keratinocytes.**

A   Representative images showing cellular localization of LL37 or sLL37 analyzed by fluorescent microscope in live primary human keratinocytes treated with FITC-labeled LL37 or sLL37 (4 μM) for 30 min followed by PBS wash. BF, bright field. Scale bar: 50 μm.

B   Representative images showing membrane localization of LL37 or sLL37 in live primary human keratinocytes treated with FITC-labeled LL37 / sLL37 and PKH26 for 30 min. Scale bar: 50 μm.

C   Immunoblotting analysis of TLR2 and pS6 in cell lysates from HaCaT keratinocytes carrying either a vector expressing a scramble shRNA or TLR2 shRNAs was exposed to LL37 (4 μM) for 2 h. pS6 protein levels were analyzed relative to total S6 (fold change).

D   Immunoblot analysis of TLR2 and pS6 in cell lysates from HaCaT keratinocytes carrying either a vector expressing TLR2 or an empty vector was exposed to LL37 (4 μM) for 2 h. pS6 protein levels were analyzed relative to total S6. Tubulin or actin was used as the loading control.

E   Primary human keratinocytes expressing TLR2 shRNAs or scramble shRNA were treated with FITC-labeled LL37 or sLL37 ± PKH26 for 30 min followed by PBS wash, and then, live cells were analyzed by fluorescent microscope. Scale bar: 50 μm.

F, G   Primary human keratinocytes were incubated with sLL37-flag or LL37-flag for 1 h. Cell lysates were immunoprecipitated with anti-flag or anti-TLR2 antibodies, showing an interaction between LL37 and TLR2. Medium, the medium of primary human keratinocytes treated with sLL37-flag or LL37-flag. All results are representative of at least three independent experiments.

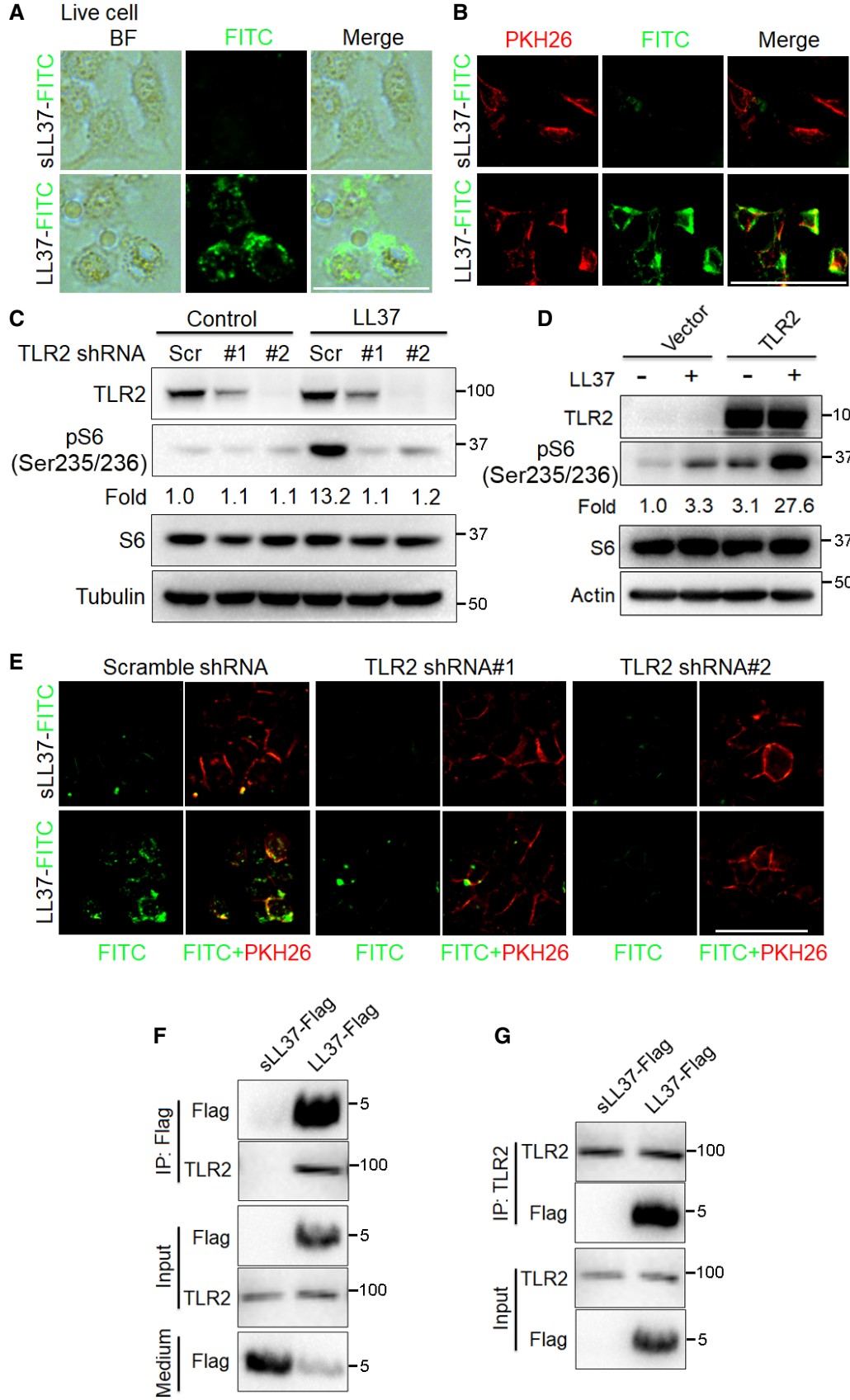

**Figure 5.**

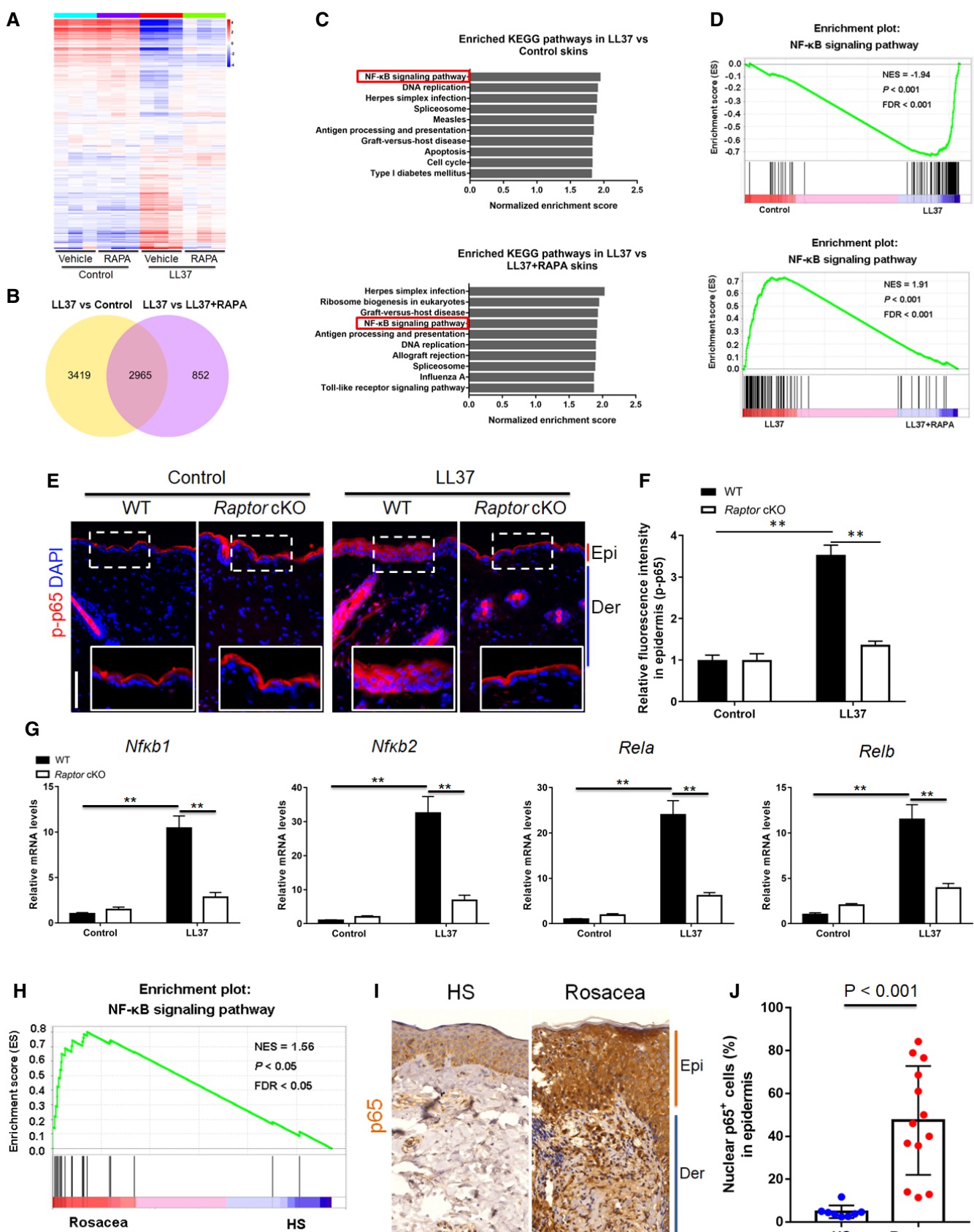

**Figure 6.**

**Figure 6. Cathelicidin LL37 stimulates NF-κB signaling via mTORC1 in keratinocytes.**

A   Heatmap of differentially regulated genes in skin lesions from LL37-injected mice administrated with RAPA or vehicle determined by RNA-sequencing (*n* = 3 independent biological samples for each group). Blue color denotes low FPKM expression; red, high FPKM expression.

B   Venn diagram showing the overlap genes between the two comparisons. LL37 vs control, LL37-injected mice administrated with vehicle vs control mice administrated with vehicle; LL37 vs LL37 + RAPA, LL37-injected mice administrated with vehicle vs LL37-injected mice administrated with RAPA.

C   Top-ranked enriched KEGG terms in genes that were differentially regulated respectively in the two comparisons (LL37 vs control, and LL37 vs LL37 + RAPA) revealed by GSEA. NF-κB signaling pathway was highlighted in red box.

D   GSEA on RNA-sequencing data from the two comparisons both shows enrichment for NF-κB signaling pathway in LL37 group. Significance was calculated by permutation test.

E   Immunostaining of phospho-p65 (p-p65) in skin sections. Bottom right panels, magnified images of dotted line areas. Epi, epidermis. Der, dermis. Scale bar: 50 μm.

F   Quantification of relative fluorescence intensity for p-p65 in epidermis (*n* = 6).

G   The mRNA expression levels of NF-κB family of transcription factors (*Nfκb1, Nfκb1, Rela,* and *Relb*) in mouse skin lesions (*n* = 6 for each group).

H   GSEA on RNA-sequencing data shows enrichment for NF-κB signaling pathway in rosacea patients. Significance was calculated by permutation test.

I   IHC of p65 on skin sections from HS and rosacea. Scale bar, 50 μm.

J   Percentage of nuclear p65-positive cells in the epidermis from HS (*n* = 8) and rosacea (*n* = 13). All results are representative of at least three independent experiments. Data represent the mean ± SEM. **$P < 0.01$. One-way ANOVA with Bonferroni's *post hoc* test (F and G) or two-tailed unpaired Student's *t*-test (J) was used.

effects observed in patients during treatment. Taken together, our pilot clinical study revealed the efficacy of topical rapamycin treatment for rosacea patients.

## Discussion

The present study reported that mTORC1 signaling is hyperactivated in rosacea. mTORC1 deletion and inhibition by its inhibitors, even with topical application, blocked the formation of rosacea-like features in rosacea-like mouse model, and hyperactivation of mTORC1 exacerbated rosacea. In keratinocyte cells, mTORC1 regulates cathelicidin via a positive feedback circuit, in which cathelicidin LL37 stimulates mTORC1 signaling through binding to TLR2. Excess cathelicidin LL37 possibly derived from this circuit induces NF-κB activation and rosacea-associated chemokines and cytokines, which may contribute to the skin inflammation of rosacea. These findings revealed an essential role of the positive feedback loop between mTORC1 signaling and cathelicidin in rosacea pathogenesis.

mTOR signaling emerges as an integrative varistor that couples cellular response to the external and internal nutrition status to dictate the inflammatory response (Weichhart *et al*, 2015). Previous studies reported that mTOR signaling was involved in multiple cutaneous diseases (Buerger *et al*, 2013; Naeem *et al*, 2017; Varshney & Saini, 2018). In this study, we revealed a positive correlation between mTORC1 signaling and the severity in rosacea patients, and deletion or inhibition of mTORC1 both blocked the development of rosacea-like skin inflammation, suggesting a key role of mTORC1 signaling in rosacea pathogenesis. TSC1 and 2 are reported

as essential upstream regulators that negatively regulate mTORC1 (Gan & DePinho, 2009). Here, we showed that TSC2 rather than TSC1 was significantly decreased in the lesional skin of rosacea, suggesting that declined TSC2 might be responsible for the hyperactivation of mTORC1 in rosacea. However, more evidence is needed to elucidate why mTORC1 is specifically hyperactivated in the epidermis of rosacea. Evidence indicated that various trigger factors (such as heat and spicy food) could initiate or exacerbate rosacea (van Zuuren, 2017; Buddenkotte & Steinhoff, 2018), but the underlying mechanisms remain largely unknown. Our findings demonstrated that heat shock and capsaicin could both stimulate mTORC1 signaling in keratinocytes, which may lay the molecular basis for these triggers in consideration of the essential role of mTORC1 in rosacea.

Recent studies on the skin barrier demonstrated that epidermis is a metabolically active construction and has adaptive functions, which may play a pivotal role in regulating inflammatory responses via activation of keratinocytes and releasing a series of proteases, cytokines and chemokines (Dainichi *et al*, 2018; Kabashima *et al*, 2019). Keratinocyte-derived proteases and cytokines might play a role in the pathogenesis of rosacea (Meyer-Hoffert & Schroder, 2011). In the present study, we surprisingly found that only ablation of epidermal mTORC1 could efficiently suppress the formation of rosacea-like skin inflammation in mice. Consistently, we also found there existed a positive correlation between mTORC1 signaling in the epidermis and the severity of rosacea patients. Taken together, these findings revealed a key role of mTORC1 signaling in epidermis in rosacea, which further highlights an important role for the inflammatory loops in epithelial microenvironment in cutaneous

**Figure 7. Cathelicidin LL37 stimulates rosacea-characteristic chemokines and cytokines via mTORC1.**

A   GSEA on RNA-sequencing data from the two comparisons (LL37 vs control, and LL37 vs LL37 + RAPA) both shows enrichment for chemokine signaling pathway in LL37 group. Significance was calculated by permutation test.

B   The mRNA expression levels of mouse chemokines (*Cxcl11, Cxcl12, Ccl2,* and *Ccl3*) in skin lesions (*n* = 6 for each group).

C   The mRNA expression levels of rosacea-associated chemokines in primary human keratinocytes treated with LL37 ± RAPA *in vitro*. Data represent the mean ± SEM (from three replicates).

D   GSEA on RNA-sequencing data from rosacea skin lesions versus HS skin samples shows enrichment for chemokine signaling pathway in rosacea. Significance was calculated by permutation test.

E   Heatmap of upregulated chemokines and cytokines in rosacea skin lesions determined by RNA-sequencing. All results are representative of at least three independent experiments. Data represent the mean ± SEM. *$P < 0.05$, **$P < 0.01$. One-way ANOVA with Bonferroni's *post hoc* test was used.

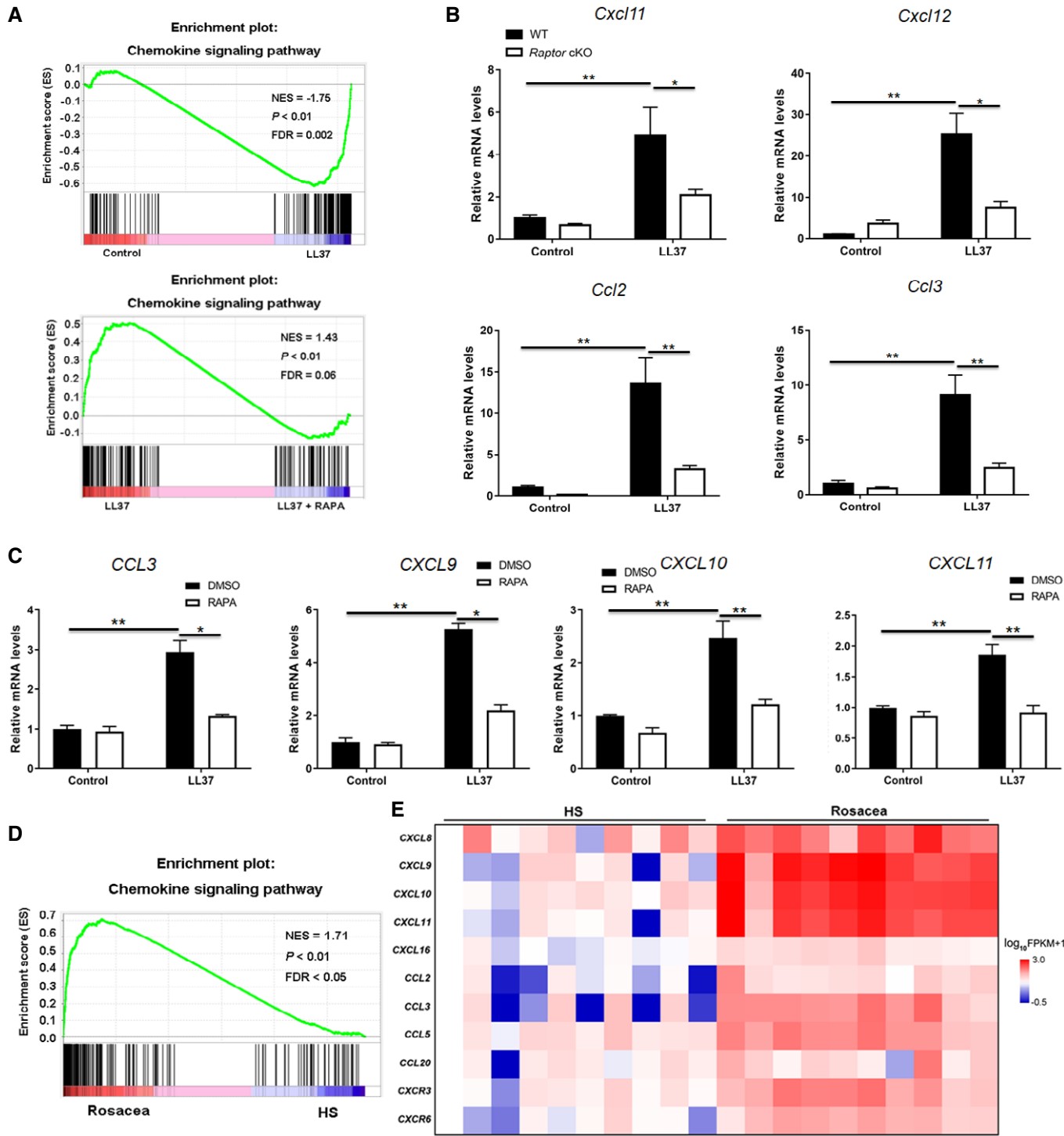

**Figure 7.**

disorders. To be mentioned, we also found mTORC1 signaling was dramatically activated in infiltrating cells in the dermis, but the specific roles in rosacea development remain unknown and need further study to figure out.

Abnormal functioning of TLR2 and cathelicidin LL37 may contribute to the dysregulation of the innate immune system and promote inflammatory cascade in rosacea (Bevins & Liu, 2007; Yamasaki

et al, 2007; Yamasaki et al, 2011). However, the connection among these aspects is still unclear. Here, we found that mTORC1 signaling is hyperactivated in rosacea and regulates cathelicidin via a positive feedback loop, in which cathelicidin LL37 stimulates mTORC1 signaling through binding to TLR2, thereby in turn promotes cathelicidin expression itself in human keratinocytes. Previous study indicated that mouse GLL34 (homologous to human LL37) played a role

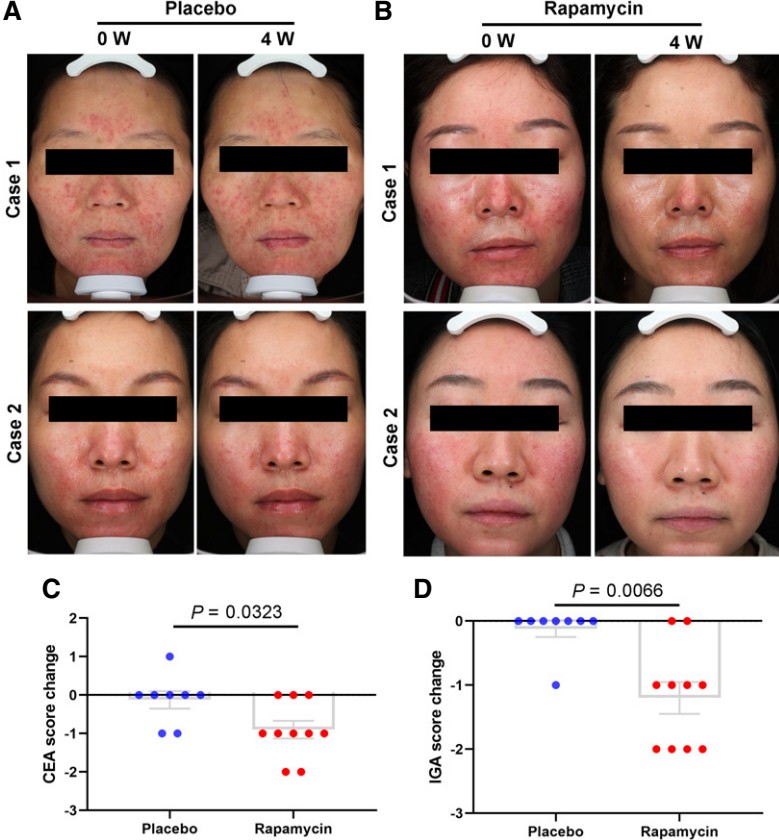

**Figure 8. Topical rapamycin treatment alleviates the rosacea-associated symptoms in rosacea patients.**

A–D Topical rapamycin treatment twice daily for 4 weeks showed an obvious therapeutic effect in rosacea patients. (A, B) Photographs of patients topically applied with rapamycin ($n = 10$) or placebo ($n = 8$) ointment at 0 week and 4th week. (C) The change of patients' CEA score showed in scatter histogram at 0 week and 4th week. (D) The change of patients' IGA score showed in scatter histogram at 0 week and 4th week. "The change value < 0" indicates that skin lesions show positive improvement after treatment; "the change value > 0" indicates that skin lesions show exacerbation after treatment; and "the change value = 0" indicates that skin lesions show no change after treatment. Data represent the mean $\pm$ SEM, and two-tailed unpaired Student's $t$-test was used (C, D).

in the development of rosacea-like skin inflammation (Yamasaki et al, 2007). In this study, we found that deficiency of mTORC1 also decreased the expression of mouse cathelicidin (CRAMP), the precursor of mouse GLL34, suggesting that mouse GLL34 might be involved in the positive feedback loop between mTORC1 signaling and LL37 in rosacea-like mouse model, and this hypothesis needs more *in vivo* data to be illustrated. To be mentioned, our results showed deficiency of mTORC1 repressed the protein level rather than mRNA level of cathelicidin, suggesting that mTORC1 regulates cathelicidin possibly at post-transcriptional level, but the precise mechanisms need to be clarified in the future. Evidence indicated that the formation of a positive feedback circuit to enhance and sustain inflammatory responses may be the mechanism by which inflammation becomes chronic (Beutler, 2009; Daniluk et al, 2012). Consistently, we demonstrated that hyperactivation of mTORC1 signaling could exacerbate and sustain the rosacea-like skin inflammation in mice. TLR2, whose well-known role is to initiate inflammatory responses by recognizing specific products of microbes or host injury, has been reported to be increased in the lesional skin of rosacea patients (Yamasaki et al, 2011; Oliveira-Nascimento et al, 2012). In this study, we provided evidence that

except microbial or host injury products, TLR2 might also recognize excess cathelicidin LL37 derived from local cells to activate mTORC1 signaling and thus trigger inflammation in rosacea. Besides TLR2, our RNA-sequencing data also showed that other TLRs (e.g., TLR4, TLR7, and TLR8) were also significantly increased in rosacea, but the role of these TLRs remains unclear in the pathogenesis of rosacea. Together, these findings suggested that the positive feedback loop between mTORC1 signaling and cathelicidin in epidermis might serve as an important regulator in elaborating the inflammatory cascades, and provide an explanation for the relapsing-remitting characteristic of rosacea.

Although rosacea is a commonly chronic inflammatory skin disorder with a relatively high prevalence, the pathophysiology remains poorly defined, leading to the fact that therapies for rosacea are focused on controlling signs and symptoms, and recurrence is common (Yamasaki et al, 2011; van Zuuren, 2017; Del et al, 2019). In the present study, we revealed an essential role for mTORC1 signaling in the pathogenesis of rosacea by employing both conditional gene deletion strategy and pharmacological inhibition method, which strongly indicates that targeting mTORC1 might be a promising option for rosacea treatment. mTORC1 inhibitor sirolimus

(also called rapamycin) has been increasingly used in dermatologic disorders, especially vascular dysfunction conditions due to its antiproliferative and antiangiogenic properties (Nadal *et al*, 2016). Topical application of rapamycin has also been tested in multiple cutaneous disorders (Leducq *et al*, 2019). Here, we found that mTOR inhibitors (including rapamycin and torin1), even topical application of rapamycin, could efficiently block the development of rosacea-like features in rosacea-like mouse model. Furthermore, our pilot clinical study showed that topical application of rapamycin significantly improved rosacea symptoms in patients, strongly suggesting the prospect of therapies targeting mTOR signaling in rosacea treatment.

NF-κB signaling has been reported to regulate various aspects of innate and adaptive functions and act as a key modulator of inflammatory responses (Liu *et al*, 2017). Previous studies demonstrated that NF-κB pathway is involved in multiple inflammatory skin disorders (Hara-Chikuma *et al*, 2015; Kang *et al*, 2016). However, the role of NF-κB signaling in the pathogenesis of rosacea is still unclear. Our previous study showed that NF-κB signaling was activated in the LL37-induced rosacea mouse model (Chen *et al*, 2019), and a recent report demonstrated that NF-κB signaling was enhanced in eyelid specimens of rosacea patients (Wladis *et al*, 2019). Consistently, our RNA-sequencing and immunochemistry data both confirmed that NF-κB signaling was strongly upregulated in the lesional skin of rosacea patients. Our data indicated that cathelicidin LL37 could induce NF-κB signaling possibly via mTORC1 signaling, and TNF-α might be the mediator by which mTORC1 activates NF-κB signaling in keratinocytes. However, the conclusions were drawn from strictly correlative data, and the precise mechanisms need more evidence to be elucidated. It is noticed that dysregulation of cytokines and chemokines is essential for the recruitment and maintenance of pathogenic inflammatory cells in target tissues, containing skin (Nedoszytko *et al*, 2014; Sidler *et al*, 2017), although the primary factors that regulate chemokine and cytokine production in specific tissues remain unclear. In rosacea lesions, a variety of cytokines and chemokines are highly expressed, indicating their important role in the pathogenesis of the disease (Gerber *et al*, 2011; Buhl *et al*, 2015). Coincided with these observations, our result demonstrated that chemokine signaling pathway is significantly enriched in the lesional skin of rosacea patients, and more importantly, we also found that cathelicidin LL37 could induce cytokines and chemokines through mTORC1 signaling in the skin. Taken together, these findings indicate that the mTORC1-NF-κB/chemokine axis plays a role in the development of rosacea.

In conclusion, the present study reveals an essential role for mTORC1 signaling in the pathogenesis of rosacea and provides a potential target for rosacea treatment.

# Materials and Methods

## Ethics statements

Human clinical studies were approved by the ethical committee of the Xiangya Hospital of Central South University. In the present study, we assure that all institutional procedures involving the use of samples and information from human volunteers were thoroughly followed. Written informed consent was obtained from every human subject. This study was registered at the Chinese Clinical Trial Registry (http://www.chictr.org.cn. ID: ChiCTR1800017380). All mice were maintained in specific pathogen-free conditions, and procedures performed were in line with instructions and permissions of the ethical committee of the Xiangya Hospital of Central South University.

## Human samples

All skin biopsies were obtained from the central face of female HS or patients with rosacea (aged 20–50 years) from the Department of Dermatology in Xiangya Hospital, Central South University. A total of 32 female rosacea patients diagnosed with rosacea by clinical and pathologic examination and a total of 28 age-matched female HS were enrolled in this study. Rosacea disease severity was assessed using CGS Score as previously described (Sanchez *et al*, 2005). The information of patients and HS is listed in Appendix Table S1. The use of all human samples was approved by the ethical committee of the Xiangya Hospital of Central South University and written informed consent was obtained from all participants, and the experiments conformed to the principles set out in the WMA Declaration of Helsinki and the Department of Health and Human Services Belmont Report.

## Mice

The BALB/c and C57BL/6 WT mice were purchased from Slack Company. *Raptor (fl/fl)*-floxed mice (Sengupta *et al*, 2010), *K14-Cre*[TM] mice (Vasioukhin *et al*, 1999), and *TSC2*[+/−] mice (Onda *et al*, 1999; Valvezan *et al*, 2017) (all C57BL/6 background) were purchased from Jackson Laboratory. *TSC2*[+/−] and *TSC2*[+/+] (WT) mice used in this study were generated by crossing *TSC2*[+/−] mice with C57BL/6 WT mice. *Raptor* cKO targeting was achieved by intraperitoneal injection of tamoxifen (150 mg/kg mice, Sigma-Aldrich) for continuous 5 days as previously described (Deng *et al*, 2015). For rapamycin treatment, BALB/c WT mice were administered with rapamycin (LC Laboratories) through intraperitoneal injection at a dose of 4 mg/kg per day as previously described (Deng *et al*, 2015) at the indicated time. For topical capsaicin treatment, 0.1% capsaicin or placebo ointment was topically applied to the BALB/c WT mice twice a day for 1 day. For topical rapamycin treatment, 0.4% rapamycin ointment was topically applied to the LL37-injected site of C57BL/6 WT mice after LL37 injection every time as previously described (Rauktys *et al*, 2008). All mice used in this study were sex-matched (male for *Raptor* cKO and *TSC2*[+/−] mouse experiments; female for BALB/c mouse experiments) at 6–8 weeks.

## LL37-induced rosacea-like mouse model

The LL37-induced rosacea-like mouse model was generated as previously described (Yamasaki *et al*, 2007). Briefly, 8-week-old WT, *Raptor* cKO, and *TSC2*[+/−] mice were shaved a day before injection and then on the back skin were intradermally injected with LL37 peptide (40 μl, 320 μM) or control vehicle twice a day for 2 days. Skin inflammation of mouse model was evaluated by the severity of erythema and edema as previously described (Chen *et al*,

2019). Briefly, the redness score was evaluated from 1 to 5, and 5 being the reddest. The redness area was measured by stereomicroscope measurements (Leica S8AP0, Leica, Germany).

## RNA-sequencing

Total RNAs of human and mouse skin samples were extracted with TRIzol Reagent (Thermo Fisher Scientific). Library preparation and transcriptome sequencing were performed using Illumina HiSeq X Ten (Novogene, Beijing, China). The mapping of 100-bp paired-end reads to genes was conducted with HTSeq v0.6.0. The fragments per kilobase of transcript per million fragments mapped (FPKM) were analyzed, and differential expression analysis was carried out using the DESeq R package (1.10.1). The hierarchical clustering heat map was produced with the ggplot library. Differential expression data were ranked through $\log_2$ fold change and performed Preranked GSEA to figure out enrichment for KEGG pathways (Subramanian et al, 2005). By adjusting the nominal P value with the Benjamini and Hochberg method, the output results were corrected for multiple comparisons.

## Histological analysis

The histological analysis was carried out as previously described (Wu et al, 2018; Zhao et al, 2018). Human and mouse skin samples were fixed in formalin and embedded in paraffin. Sections were stained with hematoxylin and eosin (H&E). The number of infiltrating cells in the dermis was determined as histological features. For counting infiltrating cells in the dermis, five areas (0.444 square inch for one) in three sections of each sample were randomly chosen, and the number of infiltrating cells in the dermis was counted.

## RT–qPCR

Total RNA was extracted from human and mouse skin tissues, and keratinocyte cells using TRIzol Reagent (Thermo Fisher Scientific), and a NanoDrop spectrophotometer (Thermo Fisher Scientific) was employed for RNA quality control. mRNA was reverse-transcribed using the Maxima H Minus First Strand cDNA Synthesis Kit with dsDNase (Thermo Fisher Scientific) following the manufacture's instructions. qPCR was conducted with iTaq™ Universal SYBR® Green Supermix (Bio-Rad) on a LightCycler 96 (Roche) thermocycler. The relative expression of each gene was assessed by using the delta CT method relative to GAPDH, and the fold change was normalized to the control group. The specific primer sequences of genes are listed in Appendix Table S2.

## Immunofluorescence

Immunofluorescence of human and mouse skin sections and cultured keratinocytes was carried out as previously described (Tang et al, 2016). Briefly, skin sections (10 μm) and keratinocytes plated on glass coverslips in 24-well plates were fixed for 10 min using 4% paraformaldehyde (PFA) and washed with phosphate-buffered saline (PBS) for three times, and then blocked for 30–60 min with blocking buffer (5% NDS, 1% BSA, 0.3% Triton X-100). Primary antibodies were incubated overnight at 4°C. Alexa Fluor 488- or 594-conjugated secondary antibody (Thermo Fisher

Scientific) was incubated for 30–60 min at room temperature. After wash, sections were counterstained with 4′,6-diamidino-2-phenylindole (DAPI). All pictures were taken with a Zeiss Axioplan 2 microscope. The fluorescence intensity was assessed with ImageJ. The following primary antibodies were used in this study: Rabbit anti-pS6 (1:200 or 1:800, Cell Signaling, catalog 4858 or 5364), Mouse anti-pS6 (1:200, Cell Signaling, catalog 62016), Rabbit anti-Raptor (1:100, Abcam, catalog ab40768), Rabbit anti-cathelicidin (1:1,000, Human Protein Atlas, catalog HPA029874), Rat anti-CRAMP (1:1,000, Novus Biologicals, catalog NB100-98689), Rabbit anti-p-Akt (ser473, catalog 4060; 1:200, Cell Signaling), Rabbit anti-p65 (1:100, Cell Signaling, catalog 8242), and Rabbit anti-p-p65 (1:200, Cell Signaling, catalog 3033).

## Immunohistochemistry

Human skin samples were fixed in formalin and embedded in paraffin, and 5-μm skin sections were cut and used. Immunohistochemistry was conducted as previously described (Deng et al, 2019). Skin sections were incubated with antibodies for pS6 (1: 400, Cell signaling, catalog 4858 or 5364) and p65 (1:400, Cell signaling, catalog 8242). As negative controls, the primary antibodies were omitted. Pictures were taken from three typical areas for each sample. The intensity of epidermal staining was evaluated as previously described (Varghese et al, 2014). Each area of images was rated on a scale of 0–4 (0 = absent staining, 1 = weak or low, 2 = moderate, 3 = strong, 4 = very strong staining).

## Cell culture and treatment

Primary human keratinocytes were isolated from human foreskin (aged 2–5) and cultured in CnT-07 (CELLnTEC, USA). HaCaT keratinocyte cell line obtained from NTCC (Biovector Science Lab, Beijing, China) was cultured in DMEM supplemented with 10% fetal bovine serum, penicillin–streptomycin, and 2 mM glutamine (Invitrogen). For heat shock treatment, keratinocyte cells were treated with heat shock (37, 42, 44, and 46°C) in the circulating water bath for the indicated time. For LL37 treatment, at a confluency of 60%, cells were starved overnight and if need incubated with rapamycin (50 nM) for 2 h, and then treated cells with LL37 (at indicated doses) for the indicated time. For sLL37- or LL37-FITC treatment, at a confluency of 40–50%, cells were starved overnight and then treated cells with sLL37- and LL37-FITC ± PKH26 for 30 min. After that, cells were washed with PBS for three times and then the images of live cells were taken with a fluorescent microscope. All experiments were performed at least three times. Routine Mycoplasma testing was performed for primary human keratinocytes and HaCaT keratinocyte cell line using LookOut Mycoplasma PCR detection (Sigma).

## Lentiviral shRNAs and plasmid

shRNAs cloned into the third-generation pLKO.1-puro vector for knocking down were purchased from Sigma-Aldrich (MISSION TRC). The Sigma clone ID for the shRNA constructs used in this study is as follows: TLR2 #1, NM_003264.2-1204s1c1; and TLR2 #2, NM_003264.2-1484s1c1. TLR2 ectopic expression vector was generated with lentivirus vector pLVX-IRES-Puro (Addgene). Lentivirus

packaging and testing were performed as previously described (Deng *et al*, 2015). HaCaT keratinocyte cells were infected with lentiviruses in growth medium containing 8 μg/ml polybrene, selected in 2 μg/ml puromycin for 7 days, followed by plating into six-well plates for subsequent treatment.

## Immunoblotting

The human skin biopsies and collected cells were lysed in RIPA buffer (Thermo Fisher Scientific) including protease inhibitors (Thermo Fisher Scientific) after washed with cold PBS. The acquired proteins were quantified via bicinchoninic acid assay (Thermo Fisher Scientific), and the prepared protein samples were separated on SDS–polyacrylamide gel electrophoresis (for immunoblotting of small peptides, such as LL37-tagged with or without flag, the samples were separated on Tricine–SDS–PAGE as previously described (Schagger, 2006)) and electroblotting to PVDF membranes. After that, the membranes were blocked with 5% non-fat milk for 60 min at room temperature and incubated with primary antibodies, and then incubated with HRP-conjugated secondary antibodies (Santa Cruz). The immunoreactive bands were visualized using the HRP substrate (Luminata, Millipore) on ChemiDoc™ XRS+ system (Bio-Rad). The following primary antibodies were used in this study: Rabbit anti-pS6 (1:2,000, Cell Signaling, catalog 4858 or 5364), Rabbit anti-S6 (1:2,000, Cell Signaling, catalog 2217), Rabbit anti-cathelicidin (1:1,000, Human Protein Atlas, catalog HPA029874), Rabbit anti-TLR2 (1:1,000, Cell Signaling, catalog 12276), Rabbit anti-p-Akt (ser473; 1:1,000, Cell Signaling, catalog 4060), Rabbit anti-Akt (1:1,000, Cell Signaling, catalog 4691), Mouse anti-mTOR (1:1,000, Proteintech, catalog 66888-1-Ig), Mouse anti-flag (1:5,000, Sigma, catalog F1804), mouse anti-β-Actin (1:5,000, Santa Cruz, catalog sc-47778), and mouse anti-α-Tubulin (1:5,000, Abcam, catalog ab7291). Data were analyzed using a GE-ImageQuant LAS 4000 mini (GE Healthcare). Quantification of pS6 and p-Akt was normalized to total S6 or Akt, respectively; quantification of other proteins was normalized to tubulin or actin by densitometry. Images have been cropped for presentation.

## Co-IP

Co-IP assays were conducted using the Dynabeads Protein G Immuno-precipitation Kit (Thermo Fisher Scientific) according to the manufacturer's instructions. Briefly, the proteins were extracted from collected fresh cells using RIPA lysis buffer containing protease and phosphatase inhibitors. A specific capture Ab recognizing the bait protein was added into the cell lysates, generating a new Ab-bait-target complex. Then, the Ab was employed as a handle to immobilize the proteins on the beads (Thermo Fisher Scientific). Proteins not immobilized were removed after wash. The complexes of bait protein were eluted from the beads and separated through boiling in SDS. Finally, the presence or absence of the target protein was determined by immunoblotting, which is the endpoint of this assay.

## ELISA

Medium from primary human keratinocytes treated with LL37 ± RAPA was centrifugated, and supernatants were plated for ELISA. ELISA kit for human TNF-α was obtained from BioLegend and performed as per manufacturer's instructions.

### The paper explained

#### Problem

Rosacea is a common chronic inflammatory skin disorder of uncertain etiology. It mainly occurs in the central face, which greatly affects the quality of life, and is associated with multiple systemic diseases (such as cancer). Although this cutaneous syndrome has been described centuries ago, its pathophysiological mechanism remains unclear. Multiple therapies have been used for the management of rosacea, including oral tetracycline and isotretinoin, topical application of azelaic acid, metronidazole, and vascular lasers. However, no specific therapeutic target has been defined, and most therapies are unsatisfactorily symptom-based treatments.

#### Results

Our study demonstrates that mTORC1 signaling is hyperactivated in the skin of rosacea patients. Ablation or specific inhibition of mTORC1 blocked the development of rosacea-like skin inflammation in a rosacea mouse model. Conversely, hyperactivation of mTORC1 signaling aggravated rosacea-like features. Mechanistically, mTORC1 signaling regulates cathelicidin in keratinocytes through a positive feedback loop, in which cathelicidin LL37 activates mTORC1 signaling by binding to Toll-like receptor 2 (TLR2), which in turn increases the expression of cathelicidin itself. Moreover, excess cathelicidin LL37 induces both NF-κB activation and disease-characteristic cytokines and chemokines via mTORC1 signaling. Importantly, topical application of rapamycin significantly improved rosacea symptoms in patients.

#### Impact

Our data suggest an essential role for mTORC1 signaling in the pathogenesis of rosacea and reveal a promising therapeutic target for rosacea treatment.

## Clinical evaluation for topical rapamycin treatment

We conducted a clinical trial in 18 adult female patients with moderate-to-severe rosacea (ETR and PPR) for 4-week topical rapamycin or placebo ointment treatment. Rosacea skin lesions were topically applied with 0.4% FDA-approved rapamycin or placebo ointment twice a day for 4 weeks as previously described (Song *et al*, 2020). All participants underwent the following evaluation at baseline and follow-up visit (at 4 weeks). Patients were evaluated by experienced dermatologists for inflammatory skin lesions with Investigator's Global Assessment (IGA) at a 5-point scale (0 = clear, 0 papules/pustules; 1 = near clear, 1–2 papules/pustules; 2 = mild, 3–10 papules/pustules; 3 = moderate, 11–19 papules/pustules; 4 = severe > 20 papules/pustules). For facial erythema, skin lesions were assessed with a Clinician's Erythema Assessment (CEA) at a 5-point scale (0 = none; 1 = mild; 2 = moderate; 3 = significant; 4 = severe) (Schaller *et al*, 2020). In the entire clinical trial, the participants were received no other drug treatment except rapamycin or placebo. Adverse events were recorded at each week.

## Statistical analysis

Statistical analysis was carried out by using SPSS 20.0 and GraphPad 7.0. Data are displayed as the mean ± SEM. We examined data for normal distribution and similar variance between groups. Statistical significance (*$P < 0.05$, **$P < 0.01$) was determined by two-tailed unpaired Student's *t*-test for comparisons

between 2 groups and 1-way analysis of variance (ANOVA) with relevant *post hoc* tests for multiple comparisons. We performed the two-tailed Mann–Whitney *U*-test for statistical analysis when the data were not normally distributed or exhibited unequal variances between the two groups. Correlation analysis was conducted by using Pearson's *r*-test or Spearman's *r*-test (for abnormally distributed data). No statistical method was employed to predetermine the size of the samples. Exact *P* values are provided in Appendix Table S4. Animals in this study were allocated randomly to different groups. The investigators were not blinded to the group allocation during data acquisition and/analysis since all samples were analyzed in the same way.

## Data availability

Sequencing data from rosacea patients have been deposited in the genome sequence archive under accession number HRA000378 (http://bigd.big.ac.cn/gsa-human/). Sequencing data for mouse model have been deposited at the GEO database, under accession number GSE147950 (https://www.ncbi.nlm.nih.gov/geo/).

**Expanded View** for this article is available online.

## Acknowledgements

We thank our colleagues (Center for Molecular Medicine, Xiangya Hospital, Central South University, China) for their generous support throughout this work. This work was supported by the National Natural Science Foundation of China (No. 81874251, No. 82073457, No. 81673086, No. 81773351, No. 81974480) and by the Science and Technology Innovation Plan of Hunan province (No. 2018SK2087).

## Author contributions

ZD and MC performed most of the experiments, analyzed the data, and wrote the manuscript. YL assisted with the establishment of LL37-induced rosacea mouse model. YL and SX assisted with molecular cloning and Co-IP experiments. YO, WS, DJ, BW, FL, JL, and QS collected the clinical samples. QP and KS assisted with immunohistochemistry experiments. YL, WX, and TL assisted with immunoblotting experiments. YZ, LS, QW, and HZ provided technical support and suggestions for the project. JL, ZD, and HX conceived the project and supervised the study. ZD and JL designed the experiments, analyzed and interpreted data, and wrote the manuscript.

## Conflict of interest

The authors declare that they have no conflict of interest.

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
