## [Review Process File · EMBO Molecular Medicine]

A positive feedback loop between mTORC1 and cathelicidin promotes skin inflammation in rosacea

Zhili Deng, Mengting Chen, Yingzi Liu, San Xu, Yuyan Ouyang, Wei Shi, Dan Jian, Ben Wang, Fangfen Liu, Jinmao Li, Qian Shi, Qinqin Peng, Ke Sha, Wenqin Xiao, Tangxiele Liu, Yiya Zhang, Hongbing Zhang, Qian Wang, Lun-Quan Sun, Hongfu Xie, and Ji Li

DOI: [10.15252/emmm.202013560](https://doi.org/10.15252/emmm.202013560)

Corresponding author: Ji Li (lijl_xy@csu.edu.cn)

Review Timeline:

Submission Date:	8th Oct 20
Editorial Decision:	30th Oct 20
Revision Received:	4th Jan 21
Editorial Decision:	1st Feb 21
Revision Received:	3rd Feb 21
Accepted:	9th Feb 21

Editor: Lise Roth

Transaction Report:

30th Oct 2020

Dear Prof. Li,

Thank you for the submission of your manuscript to EMBO Molecular Medicine. We have now received feedback from the three reviewers who agreed to evaluate your manuscript. As you will see from the reports below, the referees acknowledge the interest of the study and are overall supporting publication of your work pending appropriate revisions.

Addressing the reviewers' concerns in full will be necessary for further considering the manuscript in our journal, and acceptance of the manuscript will entail a second round of review. EMBO Molecular Medicine encourages a single round of revision only and therefore, acceptance or rejection of the manuscript will depend on the completeness of your responses included in the next, final version of the manuscript. For this reason, and to save you from any frustrations in the end, I would strongly advise against returning an incomplete revision.

When submitting your revised manuscript, please carefully review the instructions that follow below. Failure to include requested items will delay the evaluation of your revision:

- 1) A .docx formatted version of the manuscript text (including legends for main figures, EV figures and tables). Please make sure that the changes are highlighted to be clearly visible.
- 2) Individual production quality figure files as .eps, .tif, .jpg (one file per figure).
- 3) A .docx formatted letter INCLUDING the reviewers' reports and your detailed point-by-point responses to their comments. As part of the EMBO Press transparent editorial process, the point-by-point response is part of the Review Process File (RPF), which will be published alongside your paper.
- 4) A complete author checklist, which you can download from our author guidelines (<https://www.embopress.org/page/journal/17574684/authorguide#submissionofrevisions>). Please insert information in the checklist that is also reflected in the manuscript. The completed author checklist will also be part of the RPF.
- 5) Thank you for providing a Data Availability section. Please note that the datasets have to be publicly available before acceptance of the manuscript.
- 6) We would also encourage you to include the source data for figure panels that show essential data. Numerical data should be provided as individual .xls or .csv files (including a tab describing the data). For blots or microscopy, uncropped images should be submitted (using a zip archive if multiple images need to be supplied for one panel). Additional information on source data and instruction on how to label the files are available at .
- 7) Our journal encourages inclusion of *data citations in the reference list* to directly cite datasets

that were re-used and obtained from public databases. Data citations in the article text are distinct from normal bibliographical citations and should directly link to the database records from which the data can be accessed. In the main text, data citations are formatted as follows: "Data ref: Smith et al, 2001" or "Data ref: NCBI Sequence Read Archive PRJNA342805, 2017". In the Reference list, data citations must be labeled with "[DATASET]". A data reference must provide the database name, accession number/identifiers and a resolvable link to the landing page from which the data can be accessed at the end of the reference. Further instructions are available at .

8) We replaced Supplementary Information with Expanded View (EV) Figures and Tables that are collapsible/expandable online. A maximum of 5 EV Figures can be typeset. EV Figures should be cited as 'Figure EV1, Figure EV2' etc... in the text and their respective legends should be included in the main text after the legends of regular figures.

- Additional Tables/Datasets should be labeled and referred to as Table EV1, Dataset EV1, etc. Legends have to be provided in a separate tab in case of .xls files. Alternatively, the legend can be supplied as a separate text file (README) and zipped together with the Table/Dataset file. See detailed instructions here:
.

9) The paper explained: EMBO Molecular Medicine articles are accompanied by a summary of the articles to emphasize the major findings in the paper and their medical implications for the non-specialist reader. Please provide a draft summary of your article highlighting

10) For more information: There is space at the end of each article to list relevant web links for further consultation by our readers. Could you identify some relevant ones and provide such information as well? Some examples are patient associations, relevant databases, OMIM/proteins/genes links, author's websites, etc...

11) Every published paper now includes a 'Synopsis' to further enhance discoverability. Synopses are displayed on the journal webpage and are freely accessible to all readers. They include a short stand first (maximum of 300 characters, including space) as well as 2-5 one-sentences bullet points that summarizes the paper. Please write the bullet points to summarize the key NEW findings. They should be designed to be complementary to the abstract - i.e. not repeat the same text. We encourage inclusion of key acronyms and quantitative information (maximum of 30 words / bullet point). Please use the passive voice. Please attach these in a separate file or send them by email, we will incorporate them accordingly.

Please also suggest a striking image or visual abstract to illustrate your article. If you do please provide a png file 550 px-wide x 400-px high.

12) As part of the EMBO Publications transparent editorial process initiative (see our Editorial at <http://embomolmed.embopress.org/content/2/9/329>), EMBO Molecular Medicine will publish online a Review Process File (RPF) to accompany accepted manuscripts.

In the event of acceptance, this file will be published in conjunction with your paper and will include the anonymous referee reports, your point-by-point response and all pertinent correspondence relating to the manuscript. Let us know whether you agree with the publication of the RPF and as here, if you want to remove or not any figures from it prior to publication.

I look forward to receiving your revised manuscript.

Yours sincerely,

Lise Roth

Lise Roth, PhD
Editor
EMBO Molecular Medicine

***** Reviewer's comments *****

Referee #1 (Comments on Novelty/Model System for Author):

1. It is not clear what was the number of the mice (that's why I marked down technical quality)
2. Novelty is there, most of the described components have already been linked to rosacea but the role of mTOR as far as I know was unknown
3. These findings can be swiftly translated into clinical practice
4. Models are adequate

Referee #1 (Remarks for Author):

In this manuscript, Deng et al provide evidence that suggests the upregulated mTOR signaling may play a role in rosacea. Specifically, the authors suggest a mechanisms whereby cathelicidin stimulates mTOR signalling via the Toll-like receptor (TLR) pathway. In turn, it is proposed that the effects of mTOR in rosacea are mediated by NFkappaB and its target genes. To this end, it was found that the work is significant inasmuch as it links mTOR with previously recognized players in rosacea, and suggests a positive feedback loop between mTOR and cathelicidins. In my opinion, the major weaknesses of the study are the lack of mechanistic evidence explaining observed correlations between cathelicidin, mTOR and NFkappaB and insufficient dissection of the implied role of immune cells in described phenotypes. My specific comments are listed below:

Major comments:

1. Notwithstanding that correlative experiments were appreciated, it remains largely unclear how mTOR affects cathelicidin levels as well as how it activates NFkappaB (the relationship between mTOR and NFkappaB is quite complex and context dependent). Are these effects direct or indirect? What are the mediators? Are the effects of mTORC1 on NFkappaB in the context of rosacea IKK-dependent? In Fig 4G the induction of cathelicidin appears to be reversed by rapamycin. At which level does mTOR regulate cathelicidin levels?
2. It is implied that the immune system plays a major role: "Since chemokines and cytokines orchestrate inflammatory response by recruiting and activating distinct immune cells, thus induce the histopathological characteristics of rosacea (10, 14)" but this does not appear to be directly tested. I find that the authors should either directly test the role of mTOR in immune responses in skin in their model or tone down conclusions related to immune compartment.
3. Readouts in addition to phospho-rpS6 should be included when monitoring mTOR activity. This is particularly important as the Abs that were employed (at least according to the provided catalogue numbers) recognize Ser235/236 on rpS6, which can also be phosphorylated by RSKs and these residues are in fact phosphorylated even when S6K1/2 are ablated (Pende et al MCB 2004). To this end, rpS6 phosphoacceptor sites uniquely affected by mTORC1/S6K axis are Ser 240/244, and thus these pAbs should have been used. Moreover, the phosphosites should be noted in figures throughout the manuscript.
4. In multiple places number of biological replicates for mouse experiments are not indicated (Figs 1I, 2B, 2C, 2H, 2H-J, 6E). If the statement "data are representative of at least 3 independent experiments" indicates that mouse experiments were done at a N=3, this may not be sufficient to reach adequate statistical power. What was the statistical power in these experiments? Quantifications of IHC are also in large part absent.
5. Fig. 5D shows induction of phospho-rpS6 even in an apparent absence of TLR2. Was the overexpression done in a knock-out cell line? What is the explanation for the absence of TLR2 here

when compared to Scr controls from Fig. 5C.

Other comments:

1. Figs 1G/1H - Out of possible noted triggers (heat, spicy foods, UV, chemical and physical stimuli, bacteria) heat and spicy foods were chosen seemingly arbitrarily. The experiments were done only in tissue culture and not on mice. Furthermore, it is noted that capsaicin causes a dose-dependent activation of mTORC1. This may be problematic since the 100uM concentration is close to inducing cell death. In addition, blots monitoring mTOR activity should be shown pS6.
2. In order to eliminate mTORC2 only a single figure with p-Akt (Ser473) immunostaining was presented (Supp Fig 1E). Further evidence is recommended.
3. Figs 3B, 3C - Description of how quantifications were performed should be included.
4. The dynamics of phospho-rpS6 induction by LL37 in Fig. 4E are quite different from Fig. 1J. This should be commented on.
5. Fig. 5E - PKH26 labeling (as in Fig. 5B) would be helpful.
6. A number of typos were noted and thus it was thought that the article may benefit from some careful editing.

I hope that the authors will find my comments constructive and of sufficient pathos

Sincerely

I/T opisirovic

Referee #2 (Comments on Novelty/Model System for Author):

Please, look at my review! The use of HaCat cells as keratinocytes is not valid!

Referee #2 (Remarks for Author):

The manuscript by Deng et al describes a positive feedback loop between mTORC1 and cathelicidin expression, which promotes skin inflammation in rosacea, a chronic inflammatory skin disease whose pathogenesis is unclear. mTORC1 signaling is found hyper-activated in both rosacea patients and in a LL37-induced mouse model of rosacea-like skin inflammation, whereas deletion or inhibition of mTORC1 blocks the development of rosacea-like skin inflammation. The authors also show that LL37 activates mTORC1 by binding to TLR2 in HaCat cells thereby establishing a positive feedback loop via NFkB and increased cytokine expression. Furthermore, topical application of rapamycin improved the clinical symptoms in rosacea patients suggesting a novel therapeutic route for rosacea through mTORC1 inhibition.

This is an interesting paper with many in vivo data, which in principle should be suitable for publication in EMM. However, there are several major points, which should be considered in a revised version of the manuscript:

Major points:

1. The paper lacks clarity and is hard to read. Often the choice of tools e.g. individual mouse strains is not described at all. Furthermore, the logistics of experiments is unclear: human data are mixed with in vitro data of HaCat cells and various mouse models in one chapter. The manuscript should be re-written in a concise manner and the English wording must be improved.

2. One main message from the manuscript is that mTORC1 is specifically activated in rosacea, but there is no discussion what causes the specificity and whether possibly upstream-regulators such as expression of TSC1/2 are affected and being causal to upregulation of mTORC1. Along the same line, TLR-2 is specifically upregulated but not TLR4 or other receptors. What is the molecular explanation for the TLR-2 specificity?
3. Re the data analyses, most of the Figures/data are convincing. However, the data using HaCat cells in Figures 1, 5 and 7 are highly questionable. The authors refer to this aneuploid human immortalized cell line as 'keratinocytes' or 'keratinocyte cells', which is misleading and unacceptable. Some of these mechanistic data need to be repeated in primary mouse or human keratinocytes, even though these might have a limited passage number.
4. Regarding the link between mTORC1 hyperactivation and exacerbation of rosacea-like features, it is not clear whether mTOR hyperactivation without injection of LL37 is sufficient to promote skin inflammation/rosacea-like features, since there are no statistically significant differences between control and TSC2+/- mice without LL37. However, authors state that their results demonstrate that mTORC1 signaling promotes the development of rosacea.
5. Is pS6 overexpression directly correlated with an increase in mTOR1 protein levels in patients/rosacea-like mouse models? A Western blot for both proteins should be shown.
6. How do the authors reconcile their positive feedback loop with the observation that only human, but not murine LL37 injection induces rosacea-like disease in mice? This point should be discussed.

Minor points:

1. Rapamycin should be added to the list of Keywords
2. Page numbers and Figures must be labelled at the bottom of the pages
3. Are there differences in severity of rosacea features or are lesional skin areas different in different human populations? The authors only mention references that take into account the Chinese population.
4. Change "healthy individuals" to HS in Fig 6H, as HS is previously used in the manuscript.
5. Could cold/freezing temperatures affect/exacerbate rosacea? The authors mention several times that heat could exacerbate rosacea, but nothing is said about other extreme stresses e.g. cold temperatures.
6. The authors did not mention what is the blue staining in Figure 4B. The reader needs to guess that the blue color is the nuclear staining dye, which could be DAPI, Hoechst?
7. Figure 4C and Figure 6E: Red and blue colors of "EPI" and "Der" words are wrongly used for the proteins of the images, because authors used same colors.
8. Which cells produce LL37? Please, clarify if it is made in keratinocytes and/or other cells and show the data.

Referee #3 (Remarks for Author):

In this paper, Deng Z, Chen M and their colleagues found mTORC1 hyperactivation occur in both rosacea patient and LL37-induced mouse model. In a series of elegant experiments with mouse genetics and human keratinocytes studies, the authors demonstrated mTORC1 signaling is responsible for rosacea symptoms, via enhancing cathelicidin production, NFkB activation, and chemokines upregulation. Most importantly, this study provided a novel and effective clinical therapeutic method, which will be beneficial for rosacea patients. In all, I think the main conclusion from this work is based on solid experimental data, they revealed novel biology in a pathological condition, and offered potential treatment. I recommend it for publication.

Minor comments are listed below:

1. Units of length in quantification data are absent in following figure panels: Fig. 1F (pS6 in rosacea

epidermis), Fig. 2E/2J/S2E/3E (dermis-infiltrating cells).

2. Since LL37-induced rosacea model is first presented in Fig. 1, the effect of LL37 on mTORC1(Fig. 4E) and mTORC2(Fig. S4C) activation should be included in Fig. 1.

3. Cathelicidin expression should be quantified in Fig. 4G.

4. mTORC2 activation data in vivo is needed in LL37-induced mouse model.

Dear Dr. Roth:

We would like to express our deepest appreciation for your time and effort in handling our manuscript and providing the opportunity to revise our paper. To strengthen the manuscript, we carefully went through the constructive and thoughtful comments by the reviewers and the editor, and performed a series of additional experiments to fully address their concerns. The revised manuscript has now been substantially rewritten with the new data. Corresponding modifications have also been made to the figures and supplementary files. The changes in the manuscript have been highlighted in yellow color. To address the concerns raised by the reviewers, we have also prepared additional data in Supplementary file figure S8-9 (also included in the “point to point response letter”), which is for review process only.

Please find our point to point responses to the reviewers’ comments as detailed below. We are hopeful that all of the additional data, analyses and information we have provided satisfy the reviewers’ concerns.

Referee #1

Comments on Novelty/Model System for Author

Comment 1:

1. It is not clear what was the number of the mice (that's why I marked down technical quality).

Response 1:

Thank you for your careful reviewing and helpful suggestions. We apologize for not indicating the number of the mice in multiple places. In the revised manuscript, we have indicated the number of the mice used in mouse experiments, and we also detailedly answered this question in the “response to major comment 5”.

Comment 2:

2. Novelty is there, most of the described components have already been linked to rosacea but the role of mTOR as far as I know was unknown.

Response 2:

We thank the reviewer for the recognition of our work.

Comment 3:

3. These findings can be swiftly translated into clinical practice

Response 3:

We thank the reviewer for the recognition of our work, and we will keep trying to translate our findings into clinical practice.

Comment 4:

4. Models are adequate

Response 4:

We thank the reviewer for the recognition of our work.

Remarks for Author

Overall comments:

In this manuscript, Deng et al provide evidence that suggests the upregulated mTOR signaling may play a role in rosacea. Specifically, the authors suggest a mechanisms whereby cathelicidin stimulates mTOR signalling via the Toll-like receptor (TLR) pathway. In turn, it is proposed that the effects of mTOR in rosacea are mediated by NFkappaB and its target genes. To this end, it was found that the work is significant inasmuch as it links mTOR with previously recognized players in rosacea, and suggests a positive feedback loop between mTOR and cathelicidins. In my opinion, the major weaknesses of the study are the lack of mechanistic evidence explaining observed correlations between cathelicidin, mTOR and NFkappaB and insufficient dissection of the implied role of immune cells in described phenotypes. My specific comments are listed below:

Overall response:

We thank the reviewer for the recognition of our work and the insightful comments, all of which have been addressed, as detailed below.

Major comment 1:

1. Notwithstanding that correlative experiments were appreciated, it remains largely unclear how mTOR affects cathelicidin levels as well as how it activates NFkappaB (the relationship between mTOR and NFkappaB is quite complex and context dependent). Are these effects direct or indirect? What are the mediators? Are the effects of mTORC1 on NFkappaB in the context of rosacea IKK-dependent? In Fig 4G the induction of cathelicidin appears to be reversed by rapamycin. At which level does mTOR regulate cathelicidin levels?

Response 1:

We thank the reviewer for the recognition of our work and the insightful comments.

As for “How mTOR affects cathelicidin levels? And At which level does mTOR regulate cathelicidin levels?”, we first performed additional experiments to detect the mRNA expression levels in primary human keratinocytes treated with LL37 ± Rapamycin. Our results showed that the mRNA levels of cathelicidin (*CAMP*) were not affected when mTORC1 signaling was inhibited (**shown in Revised Supplementary Figure 4C**). Considering the data showing that inhibition of mTORC1 signaling significantly suppressed the increased protein levels of cathelicidin after LL37 exposure (**shown in Revised Figure 4C-G and Revised Supplementary Figure 4**), we speculate that mTORC1 may regulate cathelicidin at post-transcriptional level, which was also discussed in the Discussion part of the revised manuscript. To further figure out the mechanisms by which mTORC1 regulates cathelicidin, we conducted a series of additional experiments, including an experiment in which we knock-downed the vitamin D receptor (VDR), a major regulator of cathelicidin in epidermal keratinocytes (Segaert S. J Invest Dermatol. 2008;128:773-775), in primary human keratinocytes, then treated cells with

LL37. Our results showed that deficiency of VDR did not reverse the increased cathelicidin and mTORC1 signaling in LL37-treated keratinocytes (**shown in Supplementary file figure S8, for review only, also presented below**), suggesting that mTORC1 regulates cathelicidin in a VDR-independent manner. As for other experiments, due to the influence of COVID-19, the materials for the related experiments are relatively difficult to be obtained, resulting in a slow proceeding. However, we will continue to explore this issue, and we hope to publish the outcomes in the future.

Figure for reviewers removed

As for “How mTOR activates NF-kappaB”, we admit that the relationship between mTOR and NFkappaB might be quite complex and context dependent. To follow the reviewer’s suggestion, we also performed additional experiments to investigate the mechanism. Unexpectedly, we found that inhibition of mTORC1 signaling could reverse the upregulation of TNF- α (a typical NFkappaB activator) induced by LL37 in human primary keratinocytes *in vitro* (**shown in Revised Supplementary Figure 6D**). In the meantime, our data showed that deletion of mTORC1 in epithelial cells could decline the expression of TNF- α in the skin of rosacea mouse model (**shown in Revised Figure 2F**). Therefore, we speculate that mTORC1 might indirectly activate NFkappaB via upregulation of TNF- α , and we also discussed this issue in the Discussion part of the revised manuscript.

Major comment 2:

2. It is implied that the immune system plays a major role: "Since chemokines and cytokines orchestrate inflammatory response by recruiting and activating distinct immune cells, thus induce the histopathological characteristics of rosacea (10, 14)" but this does not appear to be directly tested. I find that the authors should either directly test the role of mTOR in immune responses in skin in their model or tone down conclusions related to immune compartment.

Response 2:

We thank the reviewer for the constructive suggestion. As suggested, we have toned down the conclusions related to immune compartment in the revised manuscript.

Major comment 3:

3. Readouts in addition to phospho-rpS6 should be included when monitoring mTOR activity. This is particularly important as the Abs that were employed (at least according to the provided catalogue numbers) recognize Ser235/236 on rpS6, which can also be phosphorylated by RSKs and these residues are in fact phosphorylated even when S6K1/2 are ablated (Pende et al MCB 2004). To this end, rpS6 phosphoacceptor sites uniquely affected by mTORC1/S6K axis are Ser 240/244, and thus these pAbs should have been used. Moreover, the phosphosites should be noted in figures throughout the manuscript.

Response 3:

We thank the reviewer for the constructive suggestion. Our *in vivo* and *in vitro* data both showed that phospho-rpS6 (Ser235/236) can be blocked by mTORC1 specific inhibitor rapamycin, suggesting that the increased phospho-rpS6 (Ser235/236) in rosacea can monitor mTORC1 activation. In the meantime, as suggested, we also used phospho-rpS6 (Ser240/244) antibody (Cell Signaling, catalog 5364) to monitor mTORC1 activity in main experiments including immunohistochemistry and immunostaining of rosacea patients and mouse model skin, and immunoblot for protein extracted from keratinocytes *in vitro* (**shown in Revised Figure 1G and H, Revised Supplementary Figure 1A-C, Revised Supplementary Figure 1K and L, and Revised Figure 4F**),

and the results of phospho-rpS6 (Ser240/244) antibody were consistent with those of phospho-rpS6 (Ser235/236). The phosphosites have been noted in revised figures throughout the revised manuscript as suggested.

Major comment 4:

In multiple places number of biological replicates for mouse experiments are not indicated (Figs 1I, 2B, 2C, S2H, 2H-J, 6E). If the statement "data are representative of at least 3 independent experiments" indicates that mouse experiments were done at a N=3, this may not be sufficient to reach adequate statistical power. What was the statistical power in these experiments? Quantifications of IHC are also in large part absent.

Response 4:

We thank the reviewer for the careful reviewing and apologize for not indicating the number of the mice in multiple places carelessly. In fact, the mouse experiments were repeated for three times, and 5-8 mice were included in each group for each time. The results of a representative mouse experiment were presented. As suggested, we also indicated the number of biological replicates for all mouse experiments in the related figure legends of revised manuscript. And the quantification of IHC was also added in the revised manuscript.

Major comment 5:

Fig. 5D shows induction of phospho-rpS6 even in an apparent absence of TLR2. Was the overexpression done in a knock-out cell line? What is the explanation for the absence of TLR2 here when compared to Scr controls from Fig. 5C.

Response 5:

We thank the reviewer for the careful reviewing. In fact, TLR2 was overexpressed in a normal cell line, not a knock-out cell line (Figure 5D). Because the overexpressing efficiency of TLR2 is quite high in keratinocyte cells, the intensity of TLR2 immunoblot band of overexpressing-cells will be much stronger than that of control cells. Therefore, when the exposure time is

suitable for the band of overexpressing-cells, the band of control cells is very weak; when the exposure time is increased, the intensity of TLR2 band of control cells will be arised (**arrow indicated**), but the intensity of band for overexpressing-cells will be strong enough to disrupt the reading of the whole result (**shown in Supplementary file figure S9, for review only, also presented below**).

Figure for reviewers removed

Other comment 1:

1. Figs 1G/1H - Out of possible noted triggers (heat, spicy foods, UV, chemical and physical stimuli, bacteria) heat and spicy foods were chosen seemingly arbitrarily. The experiments were done only in tissue culture and not on mice. Furthermore, it is noted that capsaicin causes a dose-dependent activation of mTORC1. This may be problematic since the 100uM concentration is close to inducing cell death. In addition, blots monitoring mTOR activity should be shown pS6.

Response 1:

We thank the reviewer for the insightful comments. To address the concerns, we added the pS6 (Ser240/244) to monitor mTORC1 activity in addition to pS6 (Ser235/236). First, we repeated the heatshock experiment in human primary keratinocytes, which showed that heatshock increased mTORC1 activity, which was consistent with the result in HaCaT keratinocytes (**shown in Revised Supplementary Figure 1K**). To address the concern that the increased pS6 may be problematic since the 100uM concentration is close to

inducing cell death, we detected the expression of pS6 (Ser240/244) in mouse skin topically applied with capsaicin or placebo ointment by immunostaining, demonstrating that topical application of capsaicin activates mTORC1 in epithelial cells *in vivo* (**shown in Revised Supplementary Figure 1L**).

Other comment 2:

2. In order to eliminate mTORC2 only a single figure with p-Akt (Ser473) immunostaining was presented (Supp Fig 1E). Further evidence is recommended.

Response 2:

We thank the reviewer for the constructive suggestion. As suggested, by immunostaining we detected the expression of pAkt (Ser473) in the skin of rosacea mouse model, which demonstrated that mTORC2 is not changed in rosacea-like mice skin compared to control mice (**shown in Revised Supplementary Figure 1J**). Moreover, by immunoblotting we also assessed the expression of pAkt (Ser473) in human primary keratinocytes treated with LL37. Our results showed that LL37 stimulation does not affect mTORC2 signaling in keratinocytes (**shown in Revised Figure 1H**). Based on these findings, we focused on mTORC1 rather than mTORC2 in this study.

Other comment 3:

3. Figs 3B, 3C - Description of how quantifications were performed should be included.

Response 3:

Thank you for the kind suggestion. As suggested, the description of how to quantify the redness score and area in figure 3B and C was included in materials and methods section of the revised manuscript (see in the part of “LL37-induced rosacea-like mouse model”).

Other comment 4:

4. The dynamics of phospho-rpS6 induction by LL37 in Fig. 4E are quite different from Fig. 1J. This should be commented on.

Response 4:

Fig. 1J is the immunoblotting results of protein samples from skin of LL37-induced rosacea-like mice and control mice *in vivo*, while Fig. 4E is the results of keratinocyte cells treated with different doses of LL37 *in vitro*, which might be the reason for the difference of the dynamics of phospho-rpS6 induction by LL37 between the two figures. To be mentioned, the Fig. 4E was replaced by the immunoblotting results of protein samples from primary human keratinocytes treated with different doses of LL37 *in vitro* and was provided in revised figure 1H as suggested by another reviewer. Due to space limitation, the fig. 1J was removed and replaced by fig. 1G (immunostaining of pS6 (Ser240/244) in skin of rosacea mouse model) and supplementary figure 1C (immunoblotting of pS6 (Ser240/244) for rosacea patients and healthy individuals).

Other comment 5:

5. Fig. 5E - PKH26 labeling (as in Fig. 5B) would be helpful.

Response 5:

Thank you for the helpful suggestion. As suggested, we repeated this experiment in primary human keratinocytes and PKH26 labeling was added (shown in Revised Figure 5E).

Other comment 6:

6. A number of typos were noted and thus it was thought that the article may benefit from some careful editing.

Response 6:

Thank you for the kind suggestion. We have carefully edited the whole manuscript and improved the English wording with the help from a native English speaker as suggested.

Referee #2

Comments on Novelty/Model System for Author

Comment 1:

Please, look at my review! The use of HaCat cells as keratinocytes is not valid!

Response 1:

We thank the reviewer for the constructive suggestion. As suggested, we have confirmed our main conclusions in primary human keratinocytes to fully address the reviewer's concerns, which was also detailedly answered in "response to Major comment 3".

Remarks for Author

Overall comments:

The manuscript by Deng et al describes a positive feedback loop between mTORC1 and cathelicidin expression, which promotes skin inflammation in rosacea, a chronic inflammatory skin disease whose pathogenesis is unclear. mTORC1 signaling is found hyper-activated in both rosacea patients and in a LL37-induced mouse model of rosacea-like skin inflammation, whereas deletion or inhibition of mTORC1 blocks the development of rosacea-like skin inflammation. The authors also show that LL37 activates mTORC1 by binding to TLR2 in HaCat cells thereby establishing a positive feedback loop via NFkB and increased cytokine expression. Furthermore, topical application of rapamycin improved the clinical symptoms in rosacea patients suggesting a novel therapeutic route for rosacea through mTORC1 inhibition.

This is an interesting paper with many in vivo data, which in principle should be suitable for publication in EMM. However, there are several major points, which should be considered in a revised version of the manuscript:

Overall response:

We thank the reviewer for the recognition of our work and insightful comments, all of which have been addressed, as detailed below.

Major comment 1:

1. The paper lacks clarity and is hard to read. Often the choice of tools e.g. individual mouse strains is not described at all. Furthermore, the logistics of experiments is unclear: human data are mixed with in vitro data of HaCat cells

and various mouse models in one chapter. The manuscript should be re-written in a concise manner and the English wording must be improved.

Response 1:

We thank the reviewer for the helpful suggestion. As suggested, the choice of tools e.g. individual mouse strains has been described in the material and methods of revised manuscript. We readjusted the combination of data to avoid the mix of human data with *in vitro* data of human primary keratinocytes (added as suggested) and various mouse models in one chapter. We have re-written the manuscript and improved the English wording with the help from a native English speaker as suggested.

Major comment 2:

2. One main message from the manuscript is that mTORC1 is specifically activated in rosacea, but there is no discussion what causes the specificity and whether possibly upstream-regulators such as expression of TSC1/2 are affected and being causal to upregulation of mTORC1. Along the same line, TLR-2 is specifically upregulated but not TLR4 or other receptors. What is the molecular explanation for the TLR-2 specificity?

Response 2:

We thank the reviewer for the insightful comments. As suggested, we detected the expression of TSC1 or TSC2 in rosacea. Our data demonstrated that TSC2 rather than TSC1 is significantly decreased (**shown in Revised supplementary Figure 1D**), which might be responsible for the hyperactivation of mTORC1 in rosacea. And, we discussed this issue in the discussion part of revised manuscript, but more evidence is needed to elucidate why mTORC1 is specifically hyperactivated in the epidermis of rosacea. In fact, besides TLR2, we found that TLR4 and other TLRs (eg. TLR7 and TLR8) were also increased in rosacea (**shown in Supplementary table 3**), but knockdown of these TLRs (eg. TLR4) did not reverse the activation of mTORC1 induced by LL37 in keratinocytes (data not shown). We also discussed this issue in the discussion part of revised manuscript. To be

mentioned, we are now carrying out a project to study the role of these TLRs in the pathogenesis of rosacea, and these findings will be published before long.

Major comment 3:

3. Re the data analyses, most of the Figures/data are convincing. However, the data using HaCat cells in Figures 1, 5 and 7 are highly questionable. The authors refer to this aneuploid human immortalized cell line as 'keratinocytes' or 'keratinocyte cells', which is misleading and unacceptable. Some of these mechanistic data need to be repeated in primary mouse or human keratinocytes, even though these might have a limited passage number.

Response 3:

We thank the reviewer for the helpful suggestion. As suggested, we performed additional experiments in primary human keratinocytes to confirm the main conclusions (**shown in Revised Figure 1H; Revised Figure 4E and F; Revised Figure 5A and B; Revised Figure 5E- G; Revised Figure 7C; Revised Supplementary Figure 1K; Revised Supplementary Figure 4C; Revised Supplementary Figure 6D**), with which we wish to fully address the reviewer's concerns.

Major comment 4:

4. Regarding the link between mTORC1 hyperactivation and exacerbation of rosacea-like features, it is not clear whether mTOR hyperactivation without injection of LL37 is sufficient to promote skin inflammation/rosacea-like features, since there are no statistically significant differences between control and TSC2^{+/-} mice without LL37. However, authors state that their results demonstrate that mTORC1 signaling promotes the development of rosacea.

Response 4:

We thank the reviewer for the insightful comments. Indeed, this is a very interesting question. We really did not observe rosacea-like features in TSC2^{+/-} mice without LL37 injection (the mice used for establishment of rosacea mouse model are 8 weeks old, referred to as young mice). However, we observed obvious inflammatory features (similar to rosacea-like features) in

the back skin of some TSC2+/- mice without LL37 injection when older than half a year, and the symptoms were increasingly obvious with the increasing of age. We speculate that in young TSC2+/- mice, despite the activation of mTORC1 signaling, the inflammatory responses do not break out due to the protection of homeostasis, but the homeostasis might be disrupted with age increasing, the activation of mTORC1 signaling might be enough to induce skin inflammation in old TSC2+/- mice. On the other hand, when the homeostasis in the skin is disrupted (eg. the excess expression of TLR2 or LL37), the activation of mTORC1 might exacerbate the development of rosacea.

Major comment 5:

5. Is pS6 overexpression directly correlated with an increase in mTOR1 protein levels in patients/rosacea-like mouse models? A Western blot for both proteins should be shown.

Response 5:

We thank the reviewer for the helpful suggestion. As suggested, we performed a western blot in rosacea patients and healthy individuals. Our results showed that the upregulation of pS6 was not directly correlated with an increase in mTOR protein levels (**shown in Revised Supplementary Figure 1C**).

Major comment 6:

How do the authors reconcile their positive feedback loop with the observation that only human, but not murine LL37 injection induces rosacea-like disease in mice? This point should be discussed.

Response 6:

We thank the reviewer for the insightful comments. Previous study demonstrated that the mouse GLL34 (homologous to human LL37) was increased in rosacea mouse model, and after LL37 injection the rosacea-like features was significantly alleviated in GLL34 KO mice (Camp^{-/-}) compared to WT mice (Yamasaki K, et al. Nat Med. 2007;13:975-980). Our results also showed that mouse cathelicidin (CRAMP), the precursor of GLL34, was

increased in rosacea mouse skin, and deficiency of mTORC1 could significantly reverse this increase (**shown in Revised Figure 4C and D, Revised Supplementary Figure 4A and B**). These results suggested that mouse LL37 (GLL34) might also play an important role in the formation of rosacea-like skin inflammation in LL37-induced rosacea mouse model. However, further evidence is needed to figure out this question, and we also discussed this issue in the discussion part of revised manuscript.

Minor comment 1:

1. Rapamycin should be added to the list of Keywords

Response 1:

Thank you for the kind suggestion. Rapamycin has been added to the list of Keywords in the revised manuscript as suggested.

Minor comment 2:

2. Page numbers and Figures must be labelled at the bottom of the pages.

Response 2:

As suggested, the page numbers and figures were labeled at the bottom of the pages in the revised manuscript.

Minor comment 3:

3. Are there differences in severity of rosacea features or are lesional skin areas different in different human populations? The authors only mention references that take into account the Chinese population.

Response 3:

It has been reported that humans with fair skin are more likely to be affected, whereas Asians and African Americans are less affected by rosacea (van Zuuren, E.J.. N Engl J Med, 2017. 377(18): p. 1754-1764; Steinhoff, M., et al., J Investig Dermatol Symp Proc, 2011. 15(1): p. 2-11). However, to my knowledge, there has been no evidence reporting the differences in severity of rosacea features or lesional skin areas in different human populations.

Minor comment 4:

4. Change "healthy individuals" to HS in Fig 6H, as HS is previously used in

the manuscript.

Response 4:

Thank you for the kind suggestion. We have changed "healthy individuals" to HS in Fig 6H

Minor comment 5:

5. Could cold/freezing temperatures affect/exacerbate rosacea? The authors mention several times that heat could exacerbate rosacea, but nothing is said about other extreme stresses e.g. cold temperatures.

Response 5:

To my knowledge, there has been no published evidence showing whether cold/freezing temperatures could affect/exacerbate rosacea. However, according to our clinical experience, it seems that cold/freezing temperatures can exacerbate rosacea in some patients.

Minor comment 6:

6. The authors did not mention what is the blue staining in Figure 4B. The reader needs to guess that the blue color is the nuclear staining dye, which could be DAPI, Hoechst?

Response 6:

Thank you for the kind suggestion. We have indicated the blue staining in the Revised Figure 4B as suggested.

Minor comment 7:

7. Figure 4C and Figure 6E: Red and blue colors of "EPI" and "Der" words are wrongly used for the proteins of the images, because authors used same colors.

Response 7:

Thank you for the kind suggestion. We have corrected the red and blue colors of "EPI" and "Der" words in the corresponding places of the revised figures.

Minor comment 8:

8. Which cells produce LL37? Please, clarify if it is made in keratinocytes and/or other cells and show the data.

Response 8:

Thank you for the helpful suggestion. Previous study showed that cathelicidin (the precursor protein of LL37) was mainly upregulated in the epithelial cells (keratinocytes) in rosacea, and the increased cathelicidin was proteolytically processed to generate biologically active LL37 peptide (a 37 amino acids peptide) by KLK5 (Yamasaki K, et al. Nat Med. 2007;13:975-980). Our data also showed that human cathelicidin or mouse CRAMP (homologous to human cathelicidin) was significantly increased in epithelial cells (keratinocytes) and certain inflammatory cells in rosacea patients or mouse model, which was also indicated (**shown in Revised Figure 4B and C, Revised supplementary Figure 4A**).

Referee #3

Overall comments:

In this paper, Deng Z, Chen M and their colleagues found mTORC1 hyperactivation occur in both rosacea patient and LL37-induced mouse model. In a series of elegant experiments with mouse genetics and human keratinocytes studies, the authors demonstrated mTORC1 signaling is responsible for rosacea symptoms, via enhancing cathelicidin production, NFκB activation, and chemokines upregulation. Most importantly, this study provided a novel and effective clinical therapeutic method, which will be beneficial for rosacea patients. In all, I think the main conclusion from this work is based on solid experimental data, they revealed novel biology in a pathological condition, and offered potential treatment. I recommend it for publication.

Overall response:

We thank the reviewer for the recognition of our work and helpful comments, all of which have been addressed, as detailed below.

Comment 1:

1. Units of length in quantification data are absent in following figure panels: Fig. 1F (pS6 in rosacea epidermis), Fig. 2E/2J/S2E/3E (dermis-infiltrating cells).

Response 1:

Thank you for the kind suggestion. We have added the units of length in quantification data of corresponding figure panels in the revised manuscript as suggested.

Comment 2:

2. Since LL37-induced rosacea model is first presented in Fig. 1, the effect of LL37 on mTORC1(Fig. 4E) and mTORC2(Fig. S4C) activation should be included in Fig. 1.

Response 2:

Thank you for the helpful suggestion. We have repeated the results of Fig. 4E and Fig. S4C in primary human keratinocytes suggested by another reviewer, and these figures were included in the revised figure 1H as suggested.

Comment 3:

3. Cathelicidin expression should be quantified in Fig. 4G.

Response 3:

Thank you for the kind suggestion. We have quantified the expression of cathelicidin in Fig. 4G as suggested (**shown in Revised Figure 4F**).

Comment 4:

4. mTORC2 activation data in vivo is needed in LL37-induced mouse model.

Response 4:

Thank you for the helpful suggestion. We have performed additional experiment to detected the mTORC2 activation in the skin of LL37-induced mouse model as suggested. Our results showed that mTORC2 activation was not altered in the skin of rosacea mouse model compared to control mouse (**shown in Revised Supplementary Figure 1J**).

Again, we appreciate the constructive and insightful suggestions from the

reviewers. We hope that with the revisions the manuscript will be acceptable for publication.

1st Feb 2021

Dear Prof. Li,

Thank you for the submission of your revised manuscript to EMBO Molecular Medicine. We have now received the enclosed reports from the two referees who re-reviewed your manuscript. As you will see, they are both supportive of publication, and I am therefore pleased to inform you that we will be able to accept your manuscript, once the following minor points will be addressed:

1) Referees' comments:

- Referee #1, comment on the number of mice and statistics: please address this comment in writing, and justify the number of mice used in the experiments.
- Referee #1, comment on the mechanistic understanding: if you have data at hand, we will be happy for you to include it. Alternatively, please discuss the limitations along the lines suggested by this referee.
- Referee #2: please make sure that all changes mentioned in your point-by-point letter have indeed been addressed.

2) Main manuscript text

- Please answer/correct the changes suggested by our data editors in the main manuscript file (in track changes mode). This file will be sent to you in the next few days. Please use this file for any further modification.
- Please remove the highlighted text.
- We can accommodate a maximum of 5 keywords, please adjust accordingly.
- Please reformat the references so that they are listed in alphabetical order, and with 10 authors only listed before et al.
- Please remove "data not shown". As per our guidelines on "Unpublished Data", all data referred to in the paper should be displayed in the main or Expanded View figures.
- Material and methods:
 - o Human samples: include a statement that the experiments conformed to the principles set out in the WMA Declaration of Helsinki and the Department of Health and Human Services Belmont Report.
 - o Mice: indicate the gender of the mice used in your experiments.
 - o Cells: indicate whether the cells were tested for mycoplasma contamination
- Statistics: Please indicate in the figures or in the legends the exact $n=$ and exact $p=$ values, not a range, along with the statistical test used. You may provide these values as a supplemental table in the Appendix file.
- Data Availability Section: This section should only list the new datasets generated in this study. Please note that these datasets have to be made public before acceptance of the manuscript. This section should follow the Material and Methods section. (please also see section F in the checklist)

3) Figures and Appendix:

- Please note that the Appendix Figures 2F and G are not referred to in the main text. Please make sure that all figures are referenced in the manuscript.
- Please improve the quality/resolution of all figures.
- Appendix Table S3-S8 should be renamed Table EV1-6, and a legend should be added to the respective file. The other tables should be added to the appendix files.
- Please add a table of content to the appendix, and update the names to Appendix Figure S1 etc.

- Please add/define scale bars in all Appendix figures.

4) Source Data: you submitted a file labeled Source Data and containing Figure S8 and Figure S9. Could you please clarify if these are supplemental figures or Source Data? Source data should be uploaded as 1 file per figure.

5) Checklist: Please add more information throughout, in particular for section B/1.a, sections D/8 and D/9, section E and section F (including F/20).

6) Thank you for providing The Paper Explained section. I added minor edits, please let me know if you agree with the following:

Problem

Rosacea is a common chronic inflammatory skin disorder of uncertain etiology. It mainly occurs in the central face, which greatly affects the quality of life, and is associated with multiple systemic diseases (such as cancer). Although this cutaneous syndrome has been described centuries ago, its pathophysiological mechanism remains unclear. Multiple therapies have been used for the management of rosacea, including oral tetracycline and isotretinoin, topical application of azelaic acid, metronidazole, and vascular lasers, etc. However, no specific therapeutic target has been defined, and most therapies are unsatisfactorily symptom-based treatments.

Results

Our study demonstrates that mTORC1 signaling is hyperactivated in the skin of rosacea patients. Ablation or specific inhibition of mTORC1 blocked the development of rosacea-like skin inflammation in a rosacea mouse model. Conversely, hyperactivation of mTORC1 signaling aggravated rosacea-like features. Mechanistically, mTORC1 signaling regulates cathelicidin in keratinocytes through a positive feedback loop, in which cathelicidin LL37 activates mTORC1 signaling by binding to Toll-like receptor 2 (TLR2), which in turn increases the expression of cathelicidin itself. Moreover, excess cathelicidin LL37 induces both NF- κ B activation, and disease-characteristic cytokines and chemokines via mTORC1 signaling. Importantly, topical application of rapamycin significantly improved rosacea symptoms in patients.

Impact

Our data suggest an essential role for mTORC1 signaling in the pathogenesis of rosacea and reveal a promising therapeutic target for rosacea treatment.

7) As part of the EMBO Publications transparent editorial process initiative (see our Editorial at <http://embomolmed.embopress.org/content/2/9/329>), EMBO Molecular Medicine will publish online a Review Process File (RPF) to accompany accepted manuscripts.

In the event of acceptance, this file will be published in conjunction with your paper and will include the anonymous referee reports, your point-by-point response and all pertinent correspondence relating to the manuscript. Let us know whether you agree with the publication of the RPF and as here, IF YOU WANT TO REMOVE OR NOT ANY FIGURES from it prior to publication.

I look forward to receiving your revised manuscript.

Yours sincerely,

Lise Roth

Lise Roth, PhD
Editor
EMBO Molecular Medicine

***** Reviewer's comments *****

Referee #1 (Comments on Novelty/Model System for Author):

The number of mice is 5-8 per experiments which appears to be quite low considering the effect sizes. No power information is provided (and these studies may be highly underpowered) and thus I find that the manuscript is of potentially low technical quality which is the issue that I haven't emphasized in my initial review, as the numbers of mice were missing (which I noted). Also, I thought that that the authors did not provide significant additional mechanistic evidence to support their model, and thus I ranked the novelty at "medium" as although potentially novel, I found that the proposed model was not sufficiently supported by adequate data.

Referee #1 (Remarks for Author):

Although I highly appreciate the efforts of the authors (in particular during COVID pandemic), I thought that my prior concerns regarding mechanisms that link mTOR with cathelicidin levels and NFkappaB activation were not appropriately addressed. To this end, I found that provided data are correlative and thus may not sufficiently support the authors' model. More mechanistic evidence delineating the links between mTOR, cathelicidin and NFkappaB in the context of rosacea is I think warranted, or the major conclusions should be altered to clearly indicate that they were drawn from strictly correlative data. Also, the numbers of mice seem to be quite limited (considering the effect sizes), and thus I thought that the authors should provide statistical power for animal experiments.

I hope that the authors will find my comments constructive and of sufficient pathos.

Sincerely

/Topsisirovic

Referee #2 (Remarks for Author):

The authors have addressed all my major comments. It would be nice, if some minor comments will really be changed, and not only stated as changed, but not really done....

1) Referees' comments:

Referee #1, comment on the number of mice and statistics: please address this comment in writing, and justify the number of mice used in the experiments.

Response:

The mouse experiments in this study were generally repeated for 3 times. For each replicative experiment, 5-8 mice were included for each group (not 5-8 mice per experiment, mistakenly believed by the Referee#1), meaning that there were at least 20 mice per experiment. The results of a representative mouse experiment replicate were displayed. Therefore, the statistical power for animal experiments is sufficient. And we have indicated the number of mice used in every figure suggested by data editors.

Referee #1, comment on the mechanistic understanding: if you have data at hand, we will be happy for you to include it. Alternatively, please discuss the limitations along the lines suggested by this referee.

Response:

We thank the reviewer for the constructive suggestion. For now, we have got no new data at hand for this point. As suggested, we have discussed the limitations along the lines suggested by this referee in the discussion part of the revised manuscript. In the meantime, we will continue to explore this issue, and hope to publish the outcomes in the future.

Referee #2: please make sure that all changes mentioned in your point-by-point letter have indeed been addressed.

Response:

We have carefully went through the suggestions by this referee again, and have made corresponding modifications to make sure that all changes mentioned in point-by-point letter have been addressed.

Thank you for sending the revised files. I looked at everything and all is fine. I am thus very pleased to accept your manuscript for publication in EMBO Molecular Medicine!

Please note that the GSA dataset has to be made public before publication, therefore kindly notify our editorial office as soon as this is done.

Your manuscript will then be sent to our publisher to be included in the next available issue of EMBO Molecular Medicine.

Corresponding Author Name: Ji Li

Manuscript Number: EMM-2020-13560